# RealFM: A Realistic Mechanism to Incentivize Data Contribution and Device Participation

## Abstract

Edge device participation in federating learning (FL) has been typically studied under the lens of device-server communication (*e.g.,* device dropout) and assumes an undying desire from edge devices to participate in FL. As a result, current FL frameworks are flawed when implemented in real-world settings, with many encountering the free-rider problem. In a step to push FL towards realistic settings, we propose RealFM: the first truly federated mechanism which (1) realistically models device utility, (2) incentivizes data contribution and device participation, and (3) provably removes the free-rider phenomena. RealFM does not require data sharing and allows for a non-linear relationship between model accuracy and utility, which improves the utility gained by the server and participating devices compared to non-participating devices as well as devices participating in other FL mechanisms. On real-world data, RealFM improves device and server utility, as well as data contribution, by up to 300x and 7x respectively compared to baselines.

## 1 Introduction

Federated Learning (FL) allows edge devices to collaboratively train a model under the orchestration of a central server (Centralized FL) or in a peer-to-peer manner (Decentralized FL). However, real-world implementation of FL frameworks remains stagnant due to a couple of key issues:

**(C1) Lack of Incentives**. Current FL frameworks generally lack incentives to increase device participation. This leads to training with fewer devices and data contributions, which subsequently reduces the model accuracy after federated training. Incentivizing devices to participate in training and produce more data, especially from a central server's perspective, improves model performance (further detailed in Section 3) which leads to greater benefit for the devices and server.

**(C2) Free-Rider Problem**. Second, and more importantly, many FL frameworks run the risk of encountering the free-rider problem: devices do not contribute gradient updates or data yet reap the benefits of a well-trained collaborative model. Removing the free-rider effect in FL frameworks improves performance of trained models (Wang et al., 2022; 2023) and reduces security risks (Fraboni et al., 2021; Lin et al., 2019; Wang, 2022) for devices. By simultaneously encouraging device participation while alleviating the threat of free-riding in FL frameworks, we move a step closer towards widespread adoption of FL in the real world.

In this paper, we propose RealFM: a federated mechanism $\mathcal{M}$ (*i.e.,* game) which a server in a FL setup implements to eliminate **(C1)** and **(C2)** when rational devices participate. *The goal of RealFM is to design a reward protocol, with model-accuracy $a^r$ and monetary $R$ rewards, such that rational devices choose to participate and contribute more data.* By increasing device participation and data contribution, a server trains a higher-performing model and subsequently attains greater **utility**.

Utility is the backbone of realistic FL settings. It describes the net benefit derived by devices and a server from training a model with accuracy $a$. Higher model accuracy correlates to higher utility. Every rational device $i$ aims to maximize its utility $u_i$. In realistic settings, device utility $u_i$ is personal: each device relates accuracy to utility with a unique function $\phi_i$ and has unique training costs $c_i$. Utility is the guiding factor in whether rational devices participate in a federated mechanism. Devices will only participate if the utility $u_i^r$ gained within the mechanism, via its rewards $(a_i^r, R_i)$, outstrips the maximum utility gained from local training $u_i^o$. We construct RealFM such that (i) server achieves increased utility and (ii) devices are guaranteed to receive rewards, through mechanism participation, which provide greater utility than local training $u_i^r \geq u_i^o$.

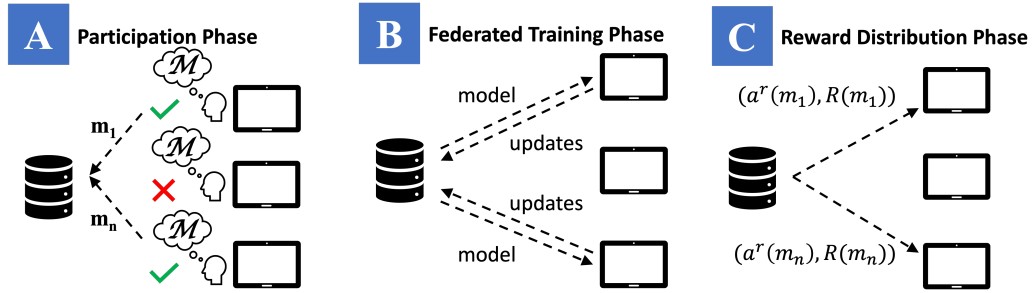

Figure 1: **Federated Mechanism Diagram. (A) Decision Phase for Device Participation**. Each device $i$, has a personalized utility $u_i^o(m)$ when training independently on $m$ local data points. Under the participation scenario, a device $i$ agrees to participate in the mechanism $\mathcal{M}$ and selects an amount of local data $m_i$ it wishes to use for model updates in FL. The quantity of data points used $m_i$ is sent to the server (no data is shared). Rational devices choose to participate if they expect the utility gained by participating in $\mathcal{M}$, $u_i^r$, is larger than their max local utility $u_i^o$. *We note that RealFM is constructed such that the utility gained by participating is **never** less than local utility, i.e., $u_i^o(m) \leq u_i^r(m), \forall m$, thereby incentivizing device participating.* **(B) Federated Training Phase**. Devices upload their updates and receive feedback from the server in an iterative manner. **(C) Accuracy & Monetary Reward Distribution Phase.** Upon completion of federated training, the server distributes both accuracy $a^r(m_i)$ and monetary $R(m_i)$ rewards to device $i$. These rewards, the crux of our proposed RealFM, serve to incentivize device participation and data contribution.

**Summary of Contributions**. We propose RealFM, a truly federated (no data sharing) mechanism. RealFM improves upon the previous federated mechanisms by,

1. provably eliminating the free-rider effect by incentivizing devices to participate and contribute more data to the federated training process than they would on their own,
2. properly modeling device utility, by allowing a non-linear relationship between accuracy and utility, thereby improving both server and device utility,
3. producing state-of-the-art results towards improving utility (for both the server and devices), data contribution, and final model accuracy on real-world datasets.

## 2 RELATED WORKS

**Federated Mechanisms.** Previous literature (Zhan et al., 2020a;b; 2021; Chen et al., 2020) have proposed mechanisms to solve **(C1)** and incentivize devices to participate in FL training. However, these works fail to address **(C2)**, the free-rider problem, and have unrealistic device utilities. In Chen et al. (2020), data sharing is allowed which is prohibitive in the FL setting, as one of the main motivations of FL is to train models in a privacy-preserving manner. In Zhan et al. (2020b; 2021; 2020a), device utility incorporates a predetermined reward for participation in federated training, with no specification on how this amount is set by the server. This is unrealistic, since rewards should be dynamic and depend upon the success (resulting model accuracy) of the federated training. Setting too low of a reward impedes device participation while too high of a reward reduces the utility gained by the central server (and risks negative utility if performance lags total reward paid out). Overall, being able to predict an optimal reward *prior* to training is unrealistic.

The recent work by Karimireddy et al. (2022) is the first to simultaneously solve **(C1)** and **(C2)**. They propose a mechanism which incentivize devices to both (i) participate in training and (ii) produce more data than on their own (data maximization). By incentivizing devices to maximize production of local data, the free-rider effect is eliminated. While this proposed mechanism is a great step forward for realistic mechanisms, pressing issues remain. First, Karimireddy et al. (2022) requires data sharing between devices and the central server. This is acceptable if portions of local data is able to be shared (*i.e.* no privacy risks exist for certain subsets of local data), yet it runs counter to the original draw of FL: a privacy-preserving training method which *does not require* sharing personal data. Second, device utility is designed in Karimireddy et al. (2022) such that *the utility improves linearly with increasing accuracy*. We find this unrealistic, as an increase in model accuracy from 49% to 50% should not hold the same utility as an increase from 98% to 99%.

**Contract Theory and Federated Free Riding.** Contract theory in FL aims to optimally determine the balance between device rewards and registration fees (cost of participation). In contract mechanisms, devices can sign a contract from the server specifying a task, reward, and registration fee. If agreed upon, the device signs and pays the registration fee. Each device receives the reward if it completes the task and receives nothing if it fails. Contract mechanisms have the ability to punish free riding in FL by creating negative incentive if a device does not perform a prescribed task (*i.e.,* it will lose its registration fee). Kang et al. (2019); Liu et al. (2022); Lim et al. (2021); Wang et al. (2021); Cong et al. (2020); Lim et al. (2020) propose such contract-based FL frameworks. While effective at improving model generalization accuracy and utility (Liu et al., 2022; Lim et al., 2020), these works focus on optimal design of rewards to devices. Our mechanism does not require registration fees, which further boosts device participation, and utilizes an accuracy shaping method to provide rewards at the end of training in an optimal and more realistic approach. Furthermore, like Karimireddy et al. (2022), our mechanism incentivizes increased contributions to federated training which is novel compared with the existing contract theory literature and mechanisms.

## 3 PRELIMINARIES

Data reigns supreme when it comes to machine learning (ML) model performance; model accuracy improves as the quantity of training data increases assuming consistency of data quality (Junqué de Fortuny et al., 2013; Zhu et al., 2012; Tramer & Boneh, 2020). Empirically, one finds that accuracy of a model is both concave and non-decreasing with respect to the amount of data used to train it (Sun et al., 2017). Training of a ML model generally adheres to the law of diminishing returns: improving model performance by training on more data is increasingly fruitless once the amount of training data is already large. Similar to Karimireddy et al. (2022), we model accuracy in terms of $m$ data points, difficulty of learning task $k > 0$, and optimal attainable accuracy $a_{opt} \in [0, 1)$,

$$a(m) := \max(\hat{a}(m), 0), \quad \hat{a}(m) := a_{opt} - \frac{\sqrt{2k(2 + \log(m/k)) + 4}}{\sqrt{m}}. \tag{1}$$

While we find empirically that $a(m)$ is both concave and non-decreasing with respect to the number of data points $m$, it is difficult to find a suitable function, such as $\hat{a}(m)$ in Equation 1, whose range is restricted to $[0, 1)$. Therefore, we allow the accuracy function $a(m) : \mathbb{Z}_0^+ \to [0, a_{opt})$ to take the form shown in Equation 1. The rationale behind this formulation is rooted in a generalization bound which can be found in Appendix B.1.

> **Assumption 1.** *Accuracy functions* $a(m) : \mathbb{Z}_0^+ \to [0, a_{opt})$ *and* $\hat{a}(m) : \mathbb{Z}^+ \to (-\infty, a_{opt})$ *are both continuous and non-decreasing.* $a(m)$ *has a root at 0:* $a(0) = 0$ *and* $\hat{a}(m)$ *is concave.*

Assumption 1 ensures that we model accuracy as we find it empirically: continuous, non-decreasing, and experiencing diminishing returns (concave) with respect to the number of data points $m$.

In the foundational federated algorithm FedAvg (McMahan et al., 2017), a server sends down a current iteration $t$ version of the parameters $w^t$. Devices perform $h$ local gradient updates (using their own data) on $w^t$ before sending the result to a central server which aggregates all local updates together into $w^{t+1}$. This process is repeated until an optimum or a near optimum is reached. Our paper follows this same FL setting, with $n$ devices collaboratively train the parameters $w$ of a ML model. Each device has its own local dataset $\mathcal{D}_i$ (which can change in size) containing data independently chosen from a common distribution D (known as the IID setting). Unlike FedAvg and many FL methods, we *do not require* devices participate in training. Instead, devices collaborate with a central server only when they deem it beneficial for their own utility (*i.e.*, collaboration with the central server results in a model with higher accuracy or monetary rewards). In the FL setting, we redefine $m$ to become $m_i := |\mathcal{D}_i|$ for a device $i$ and $m := \sum_j |\mathcal{D}_j|$ for all participating devices.

## 4 REALISTIC DEVICE UTILITY

RealFM improves upon previous federated mechanisms, such as Karimireddy et al. (2022); Zhan et al. (2021; 2020b), by modeling utility in a more realistic manner. Specifically, we design a utility which: **(D1) Realistically Relates Accuracy to Utility (D2) Allows Non-Deterministic Rewards**.

To tackle **(D1)**, we propose utilizing an accuracy payoff function $\phi_i(a) : [0, 1) \rightarrow \mathbb{R}_{\geq 0}$, which allows for a flexible definition of the benefit device $i$ receives from having a model with accuracy $a$. In Karimireddy et al. (2022), it is assumed that $\phi_i$ is linear, $\phi_i(a) = a$, for all devices, which may not hold in many instances. Non-linearity is necessary to precisely relate accuracy to utility, however, as accuracy improvement from 49% to 50% should be rewarded much differently than 98% to 99%. Therefore, we generalize $\phi_i$ to become a convex and increasing function (which includes the linear case). The convex and increasing requirements stems from the idea that increasing accuracy leads to increasing benefit for rational devices.

> **Assumption 2.** *Each device $i$'s accuracy payoff function $\phi_i(a(m)) : [0, a_{opt}) \rightarrow \mathbb{R}_{\geq 0}$ is continuous, non-decreasing, and has a root at 0: $\phi_i(0) = 0$. Furthermore, $\phi_i(a)$ is convex and strictly increasing with respect to $a$, while $\phi_i(a(m))$ is concave and strictly increasing $\forall m$ such that $\hat{a}(m) \geq 0$.*

Many realistic choices for $\phi_i$ which satisfy Assumption 2 exist. One reasonable choice is $\phi_i(a) = \frac{1}{(1-a)^2} - 1$. This choice of $\phi_i$ more precisely models how utility should grow as accuracy approaches 100%, especially when compared to the linear relationship $\phi_i(a) = a$.

## 4.1 DEFINING SERVER UTILITY

*The overarching goal for a central server $C$ is to attain a high-performing ML model resulting from federated training.* As discussed in Section 3, ML model accuracy is assumed to improve as the total amount of data contributions $m$ increases. Therefore, server utility $u_C$ is a function of model accuracy which in turn is a function of data contributions,

$$u_C(m) := p_m \cdot \phi_C\left(a\left(\sum m\right)\right). \quad (2)$$

The parameter $p_m \in (0, 1]$ denotes the central server's profit margin (percentage of utility kept by the central server). Server utility in Equation 2 is maximized when model accuracy approaches 100%, which in turn occurs when $\sum m \rightarrow \infty$. The more data contributed by all devices $m$ to the server, the better a ML model performs. The better a ML model performs, the greater server utility becomes. Thus, an optimal mechanism $\mathcal{M}$ from the server's standpoint *would be one which incentivizes data contributions.*

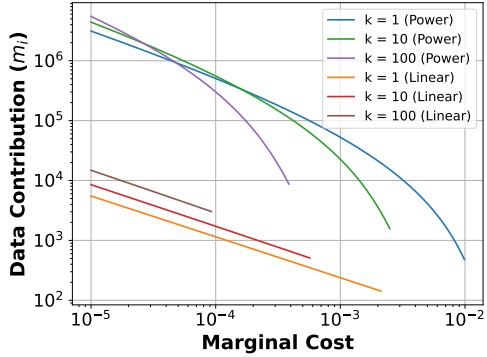

Figure 2: **Local Optimal Data Contribution for Varying Payoff Functions.** We compare how optimal data contribution varies when different payoff functions are used. As expected, a more realistic power payoff function, $\phi_i(a) = \frac{1}{(1-a)^2} - 1$, results in a greater optimal data contribution compared to a linear payoff function, $\phi_i(a) = a$. We defined $\hat{a}(m)$ as in Equation 1, with $a_{opt} = 0.95$ and multiple values of $k$.

Through the use of the profit margin $p_m$ in Equation 2, we are able to tackle **(D2)** and allow for non-deterministic rewards. Only a $p_m$ fraction of the total collected utility is kept by the server in Equation 2. The remaining $(1 - p_m)$ utility can be distributed back to participating devices in any manner. In Section 5.1, we detail how RealFM distributes the rewards proportionally to how much data each participating device contributes.

## 4.2 DEFINING LOCAL DEVICE UTILITY

While increasing the amount of training data improves the performance of a model, it can be expensive for devices to collect more data. Each device $i$ has its own fixed marginal cost $c_i > 0$, which represents the cost for collecting and computing an extra data point. For example, the cost incurred for collecting $m$ data points would be $c_i m$. Overall, a device must weigh the cost incurred for collecting more data versus the benefits obtained from having a more accurate model. We model this relationship through the use of a utility function $u_i$ for each device $i$. Mathematically, the utility for each device $i$ is defined as,

$$u_i(m) = \phi_i(a(m)) - c_i m. \quad (3)$$

The utility $u_i$ for each device $i$ is a function of data contribution: devices determine how many data points $m$ to collect in order to balance the benefit of model accuracy $\phi_i(a(m))$ versus the costs of data collection $-c_i m$.

> **Theorem 1** (Optimal Local Data Collection). *Consider device $i$ with marginal cost $c_i$, accuracy function $\hat{a}(m)$ satisfying Assumption 1, and accuracy payoff $\phi_i$ satisfying Assumption 2. Device $i$ collects an optimal amount of data $m_i^o$,*
>
> $$m_i^o = \begin{cases} 0 & \text{if } \max_{m_i \geq 0} u_i(m_i) \leq 0, \\ m^*, \text{ such that } \phi_i'(\hat{a}(m^*)) \cdot \hat{a}'(m^*) = c_i & \text{else.} \end{cases} \quad (4)$$

Theorem 1 details how much data a device collects when *training on its own and thereby not participating in a federated training scheme*. Figure 6 showcases utility peaks for both non-linear and linear accuracy payoff functions. Figure 6c shows that utility peaks at a negative value when marginal costs become too large. In this case, as shown in Theorem 1, devices do not contribute data. We defer proof of Theorem 1 to Appendix B.

## 5 REALISTIC FEDERATED MECHANISMS

In the standard FL setting, devices are *expected* to participate in training, i.e., send gradients or locally updated models to the central server. However, this is not a realistic assumption in practice. Many devices choose to participate in federated training only when it provides benefit, or utility, to them. In the eyes of rational devices, they deserve adequate reward for shouldering computational burdens and providing valuable new gradient contributions.

To entice devices to participate in federated training, a central server must incentivize them. Our paper covers two realistic rewards that a central server can provide: **(i)** model accuracy and **(ii)** monetary. The interaction between the server and devices is formalized by a *mechanism* $\mathcal{M}$. When participating in the mechanism, a device $i$ performs federated training for a global model $\boldsymbol{w}$ on $m_i$ local data points in exchange for model accuracy $a^r(m_i) \in [0, a_{opt})$ and monetary $R(m_i) \in \mathbb{R}_{\geq 0}$ rewards. This exchange is defined as the following,

$$\mathcal{M}(m_1, \cdots, m_n) = ((a^r(m_1), R(m_1)), \cdots, (a^r(m_n), R(m_n))). \quad (5)$$

> **Definition 1** (Feasible Mechanism). *A feasible mechanism $\mathcal{M}$ (1) returns a bounded accuracy $0 \leq a^r(m_i) \leq a(\sum \boldsymbol{m})$ and non-negative reward $R(m_i) \geq 0$ to device $i$ and (2) is bounded in its provided utility $0 \leq u_i^r(\boldsymbol{m}) \leq u_{max}^r$.*

> **Definition 2** (Individual Rationality). *Let all devices contribute $\boldsymbol{m}$ data points and have $\boldsymbol{c}$ marginal costs. Mechanism $\mathcal{M}$ provides utility $u_i^r$ via model-accuracy $a^r(m_i)$ and monetary $R(m_i)$ rewards distributed to each device $i$. A mechanism satisfies individual rationality if for any device $i \in [n]$ (with maximum local utility $u_i^o$) and contribution $\boldsymbol{m}$,*
>
> $$\phi_i(a^r(m_i)) + R(m_i) \geq \phi_i(a(m_i)) \equiv u_i^r \geq u_i^o. \quad (6)$$

In order to incentivize devices to participate in a FL mechanism, it must (i) return a feasible utility $[\mathcal{M}^U(\cdot)]_i$ (fulfilling Definition 1) and (ii) provide utility that is greater than or equal to the utility gained by not participating and training alone (fulfilling Definition 2).

> **Theorem 2** (Existence of Pure Equilibrium). *Consider a feasible mechanism $\mathcal{M}$ returning utility $[\mathcal{M}^U(m_i; \boldsymbol{m}_{-i})]_i$ to device $i$ ($[\mathcal{M}^U(0; \boldsymbol{m}_{-i})]_i = 0$). Participating device $i$'s utility is,*
>
> $$u_i^r(m_i; \boldsymbol{m}_{-i}) := [\mathcal{M}^U(m_i; \boldsymbol{m}_{-i})]_i - c_i m_i. \quad (7)$$
>
> *If $u_i^r(m_i, \boldsymbol{m}_{-i})$, is quasi-concave for $m_i \geq m_i^u := \inf\{m_i \mid [\mathcal{M}^U(m_i; \boldsymbol{m}_{-i})]_i > 0\}$ and continuous in $\boldsymbol{m}_{-i}$, then a pure Nash equilibrium with $\boldsymbol{m}^{eq}$ data contributions exists,*
>
> $$u_i^r(\boldsymbol{m}^{eq}) = [\mathcal{M}^U(\boldsymbol{m}^{eq})]_i - c_i \boldsymbol{m}_i^{eq} \geq [\mathcal{M}^U(m_i; \boldsymbol{m}_{-i}^{eq})]_i - c_i m_i \; \forall m_i \geq 0. \quad (8)$$

The proof of Theorem 2 is found in Appendix B. Our new proof amends and simplifies that of Karimireddy et al. (2022), as we show $u_i^r$ only needs to be quasi-concave in $m_i$.

---

**Algorithm 1:** RealFM: A Realistic Federated Mechanism

---

**Input:** Data contributions $\boldsymbol{m}$, marginal costs $\boldsymbol{c}$, payoff functions $\boldsymbol{\phi}$, profit margin $p_m$, $h$ local steps, $T$ total iterations, total local epochs $E$, initial model parameters $\boldsymbol{w^1}$, loss $\ell$, step-size $\eta$, accuracy-shaping function $\gamma_i(m)$, and accuracy function $a(m)$.
**Output:** Model with accuracy $a^r(m_i)$ and monetary reward $R(m_i)$.

1   $s_i \leftarrow m_i / \sum_{j=1}^{n} m_j$      // compute self-weight for federated averaging
2   **for** $t = 1, \ldots, T$ **do**
3      Server distributes $\boldsymbol{w^t}$ to all devices
4      **for** $h$ local steps, each device $i$ **in parallel do**
5          $\boldsymbol{w_i^{t+1}} \leftarrow \text{ClientUpdate}(i, \boldsymbol{w^t})$
6      $\boldsymbol{w^{t+1}} \leftarrow \sum_j s_j \boldsymbol{w_i^{t+1}}$

7   $r(\boldsymbol{m}) \leftarrow (1 - p_m) \dfrac{\phi_C\left(a\left(\sum_j m_j\right)\right)}{\sum_j m_j}$          // compute monetary reward

8   **for** $i = 1$ *to* $n$ **do**
9      Compute $m_i^o$ and $m_i^*$ using $c_i, \phi_i$, and $\gamma_i(m_i)$
10     Return $(a^r(m_i), R(m_i))$ to device $i$ using $m_i, m_i^o, m_i^*, a(m), \gamma_i(m_i), \phi_i, r(\boldsymbol{m})$ in Eq. 13

11   ClientUpdate($i, \boldsymbol{w}$):
12   $\mathcal{B} \leftarrow$ batch $m_i$ data points
13   **for** *each epoch* $e = 1, \ldots, E$ **do**
14     **for** *batch* $b \in \mathcal{B}$ **do**
15        $\boldsymbol{w} \leftarrow \boldsymbol{w} - \eta \nabla \ell(\boldsymbol{w}; b)$

---

## 5.1   DEFINING MECHANISM REWARDS

While the mechanism in Karimireddy et al. (2022) was the first federated mechanism proposed to maximize data collection, it required data sharing and lacked realistic assumptions on device utility. We design a realistic and truly federated mechanism which incentivizes device participation and data contribution by returning **(i)** monetary rewards to devices when desirable and **(ii)** improved accuracy, via accuracy-shaping, under non-linear and non-uniform accuracy payoff functions $\phi_i$.

**Monetary Rewards**. As mentioned above in Section 5, the federated mechanism $\mathcal{M}$ is posted by the central server. The profit margin $p_m$ of the server is fixed and known by all devices when $\mathcal{M}$ is issued. However, the accuracy payoff function for the central server $\phi_C$ is not known by the devices.

Section 4.1 details how the server maintains a $p_m$ fraction of utility it receives, as shown in Equation 2, while distributing the remaining $(1 - p_m)$ utility amongst participating devices as a monetary reward. The reward is dispersed as a marginal monetary reward $r(\boldsymbol{m})$ per contributed data point,

$$r(\boldsymbol{m}) := (1 - p_m) \cdot \phi_C\left(a\left(\sum \boldsymbol{m}\right)\right) / \sum \boldsymbol{m}. \tag{9}$$

The marginal monetary reward $r(\boldsymbol{m})$ is dynamic and depends upon the total amount of data contributions from devices during federated training. Therefore, $r(\boldsymbol{m})$ is unknown by devices when $\mathcal{M}$ is issued. However, the server computes and provides the monetary rewards once training is complete.

**Accuracy Shaping**. Accuracy shaping, the heart of any great federated mechanism, is responsible for incentivizing devices to collect more data than their local optimal amount $m_i^o$. The idea behind accuracy shaping is to incentivize device $i$ to produce more data $m_i^* \geq m_i^o$ by providing it a boosted model accuracy whose utility outstrips the marginal cost of collecting more data $-c_i \left(m_i^* - m_i^o\right)$.

Our mechanism is related to Karimireddy et al. (2022) in that it also performs accuracy shaping, albeit in a more difficult setting. Accuracy shaping with non-linear accuracy payoffs $\phi$ is far more difficult than with linear ones (Karimireddy et al. (2022) assumes a linear $\phi$). To overcome the issues with non-linear $\phi$'s, we carefully construct an accuracy-shaping function $\gamma_i(m)$ for each device $i$.

**Theorem 3** (Accuracy Shaping Guarantees). *Consider a device $i$ with marginal cost $c_i$ and accuracy payoff function $\phi_i$ satisfying Assumption 2. Denote device $i$'s optimal local data contribution as $m_i^o$ and its subsequent accuracy $\bar{a} = a(m_i^o)$. Define the derivative of $\phi_i(a)$ with respect to $a$ as $\phi_i'(a)$. For any $\epsilon \to 0^+$ and marginal server reward $r(\boldsymbol{m}) \geq 0$, device $i$*

*has the following accuracy-shaping function $\gamma_i(m)$ for $m \geq m_i^o$,*

$$\gamma_i := \begin{cases} \frac{-\phi_i'(\bar{a}) + \sqrt{\phi_i'(\bar{a})^2 + 2\phi_i''(\bar{a})(c_i - r(\boldsymbol{m}) + \epsilon)(m - m_i^o)}}{\phi_i''(\bar{a})} & \text{if } \phi_i''(\bar{a}) > 0 \\ \frac{(c_i - r(\boldsymbol{m}) + \epsilon)(m - m_i^o)}{\phi_i'(\bar{a})} & \text{if } \phi_i''(\bar{a}) = 0 \end{cases} \quad (10)$$

*Given the defined $\gamma_i(m)$, the following inequality is satisfied for $m \in [m_i^o, m_i^*]$,*

$$\phi_i(a(m_i^o) + \gamma_i(m)) - \phi_i(a(m_i^o)) > (c_i - r(\boldsymbol{m}))(m - m_i^o). \quad (11)$$

*The new optimal contribution is $m_i^* := \{m \geq m_i^o \mid a(m + \sum_{j \neq i} m_j) = a(m_i^o) + \gamma_i(m)\}$. Accuracy shaping improves device $i$'s data contribution $m_i^* \geq m_i^o$ for any contribution $\boldsymbol{m}_{-i}$.*

Theorem 3, whose full proof is found in Appendix B, details the limits of accuracy shaping such that any mechanism can remain feasible. To remain feasible, the central server cannot provide provide an accuracy beyond the sum of the total contributions $a(\sum \boldsymbol{m})$. Devices which participate in $\mathcal{M}$ must share $c_i, \phi_i$ with the central server while the central server must provide $p_m$ when posting $\mathcal{M}$.

**Remark 1.** *For a linear accuracy payoff, $\phi_i(a) = wa$ for $w > 0$, and thus $\phi_i''(a) = 0 \; \forall a$, Equation 10 relays $\gamma_i = \frac{(c_i - r(\boldsymbol{m}) + \epsilon)(m - m_i^o)}{w}$. We recover the accuracy-shaping function of Equation (13) in Karimireddy et al. (2022) with their no-reward $r(\boldsymbol{m}) = 0$ and $w = 1$ linear setting. Thus, our accuracy-shaping function generalizes the one in Karimireddy et al. (2022).*

## 5.2 REALFM: A REALISTIC FEDERATED MECHANISM

Pseudo-code for our mechanism, RealFM, is detailed in Algorithm 1. We begin by first defining the *utility* $[\mathcal{M}^U(\boldsymbol{m})]_i$ that our mechanism returns to each device $i$,

$$[\mathcal{M}^U(\boldsymbol{m})]_i = \begin{cases} \phi_i(a(m_i)) & \text{if } m_i \leq m_i^o \\ \phi_i(a(m_i^o) + \gamma_i(m_i)) + r(\boldsymbol{m})(m_i - m_i^o) & \text{if } m_i \in [m_i^o, m_i^*] \\ \phi_i(a(\sum \boldsymbol{m})) + r(\boldsymbol{m})(m_i - m_i^*) & \text{if } m_i \geq m_i^*. \end{cases} \quad (12)$$

Now we mathematically formulate how $\mathcal{M}$ distributes model-accuracy and monetary rewards to each device participating in $\mathcal{M}$ given a collective data contribution $\boldsymbol{m}$,

$$\left(a^r(m_i), R(m_i)\right) = \begin{cases} \left(a(m_i),\; 0\right) & \text{if } m_i \leq m_i^o \\ \left(a(m_i^o) + \gamma_i(m_i),\; r(\boldsymbol{m})(m_i - m_i^o)\right) & \text{if } m_i \in [m_i^o, m_i^*] \\ \left(a(\sum \boldsymbol{m}),\; r(\boldsymbol{m})(m_i - m_i^*)\right) & \text{if } m_i \geq m_i^*. \end{cases} \quad (13)$$

**Theorem 4** (Existence of Equilibrium with Increased Data Contribution). *Our realistic federated mechanism $\mathcal{M}$, defined in Equation 13, performs accuracy-shaping with $\gamma_i$ defined in Theorem 3 for each device $i \in [n]$ and some small $\epsilon \to 0^+$. As such, $\mathcal{M}$ is Individually Rational (IR) and has a unique Nash equilibrium at which device $i$ will contribute $m_i^* \geq m_i^o$ updates, thereby eliminating the free-rider phenomena. Furthermore, since $\mathcal{M}$ is IR, devices are incentivized to participate as they gain equal to or more utility than by not participating.*

**Corollary 1.** *When accuracy payoffs are linear for a device $i$, $\mathcal{M}$ defined in Equation 13 is contribution-maximizing. The proof follows from Theorem 4.2 in Karimireddy et al. (2022).*

Our usage of a non-linear accuracy payoff function $\phi_i$ promotes increased data production compared to a linear payoff (see Section 6). However, proof of data maximization for non-linear payoffs is not possible as one cannot perfectly bound the composition function $\phi(a + \gamma(m))$ as you can with the linear payoff $a + \gamma(m)$. Our accuracy-shaping function is still data maximizing when linear.

**Honest Devices.** Our setting assumes honest devices. They only store the rewarded model returned by the server after training. In the case that devices are dishonest, slight alterations to the federated training scheme can be made. Namely, schemes that train on varying-sized models such as (Diao et al., 2020; Li & Wang, 2019; Hu et al., 2021; Arivazhagan et al., 2019) can be implemented (where model sizes correspond to the amount of data contributed) to alleviate honesty issues.

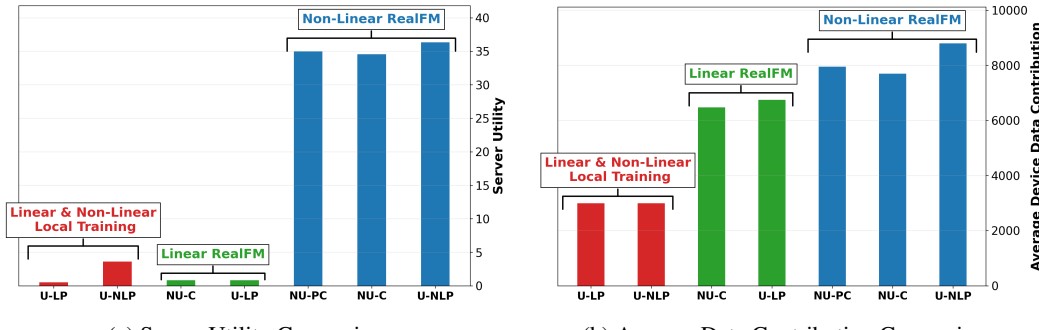

(a) Server Utility Comparison.  (b) Average Data Contribution Comparison.

Figure 3: **Improved Server Utility on CIFAR-10.** RealFM increases central server utility (3a) on CIFAR-10 for 16 devices compared to relevant baselines. Tasked with non-linear (and possibly non-uniform) payoff functions, RealFM achieves over 15x more utility than a FL version of (Karimireddy et al., 2022) (denoted as Linear RealFM). RealFM incentivizes devices to produce over 250% more data points (3b) than they would on their own.

## 6 EXPERIMENTAL RESULTS

To test the efficacy of RealFM, we analyze how well it performs at **(i)** improving utility for the central server and devices, and **(ii)** increasing the amount of data contributions to the federated training process on image classification experiments. We use CIFAR-10 (Krizhevsky et al., 2009) and MNIST (Deng, 2012) as our real-world datasets.

**Experimental Baselines.** Few FL mechanisms eliminate the free-rider effect, with none doing so without sharing data, in the non-convex setting. Therefore, we adapt the mechanism proposed in (Karimireddy et al., 2022) as the baseline to compare against (we denote this adaptation as Linear RealFM). We also compare RealFM to a local training baseline where we measure the average device utility attained by devices if they did not participate in the mechanism. Server utility is inferred in this instance by using the average accuracy of locally trained models.

**Testing Scenarios.** We denote (LP) as linear payoff and (NLP) as non-linear payoff. We test RealFM under uniform (U-NLP) as well as non-uniform costs (NU-C) and payoffs & costs (NU-PC). Since Linear RealFM does not use a payoff function, we only test it under uniform (U-LP) and non-uniform costs (NU-C). Finally, for local training, the non-uniform results are nearly identical to their uniform counterparts, so we remove them to save space.

**Experimental Setup.** Within our experiments, both 8 and 16 devices train a ResNet18 and a small convolutional neural network for CIFAR-10 and MNIST respectively. As detailed in Appendix C, we carefully select and tune $a(m)$ to match the empirical training results on both datasets as closely as possible. Once tuned, we select a marginal cost $c_e$ such that, by Equation 4, the local optimal data contribution is $3,000$ and $5,500$ ($3,500$ and $7,000$) for each of the 8 and 16 devices respectively on CIFAR-10 (MNIST). When performing uniform cost experiments, each device uses $c_e$ as its marginal cost (and thus each device has the same amount of data). For non-uniform cost experiments, $c_e$ is the mean of a Gaussian distribution from which the marginal costs are sampled. Our uniform payoff function is $\phi_i(a) = \frac{1}{(1-a)^2} - 1$ for each device $i$. For non-uniform payoff experiments, we set $\phi_i(a) = z_i\left(\frac{1}{(1-a)^2} - 1\right)$ where $z_i$ is uniformly sampled within $[0.9, 1.1]$. A list of hyper-parameters, additional experimental details, and 8 devices results are found in Appendix C.

**RealFM Dramatically Improves Server Utility.** RealFM is powered by the incentives provided by its accuracy-shaping function. By Theorem 3, devices provably contribute more data when accuracy-shaping occurs. As the total amount of data $m$ grows due to the incentives of accuracy shaping, a higher-performing model is trained by the server. A higher-performing model results in higher utility for the server (Equation 2). This is backed up empirically, as Figures 3a and 4a showcase upwards of *3 magnitudes greater utility* compared to state-of-the-art FL mechanisms.

**RealFM Incentivizes Devices to More Actively Contribute to Federated Training.** Figures 3b and 4b showcase the power of RealFM, via its accuracy-shaping function, to incentivizes devices to contribute more data points. Through its construction, RealFM's accuracy-shaping function incentivizes devices to contribute more data than they would locally in exchange for greater utility.

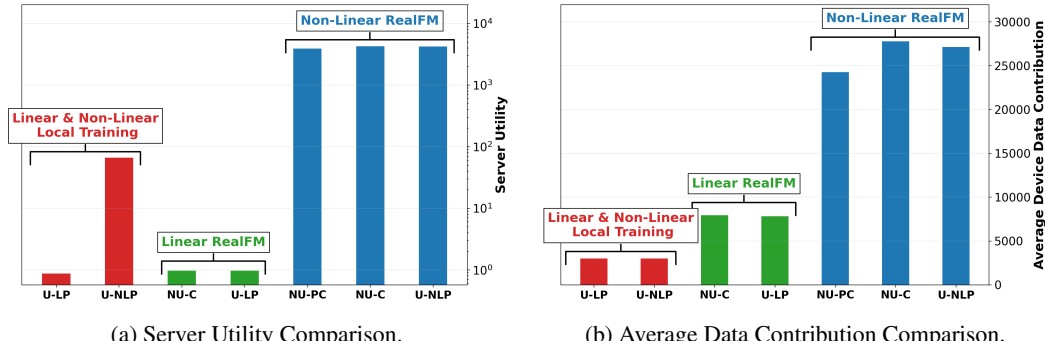

(a) Server Utility Comparison.

(b) Average Data Contribution Comparison.

Figure 4: **Improved Server Utility on MNIST.** Tasked with non-linear (and possibly non-uniform) payoff functions, RealFM achieves *over 3 magnitudes* more server utility than a FL version of (Karimireddy et al., 2022) (denoted as Linear RealFM). RealFM incentivizes devices to produce nearly 700% more data points (4b) than on their own.

This is important for two reasons. First, incentivizing devices to contribute more than local training proves that the free-rider effect is not taking place. Second, higher data contribution lead to better-performing models and higher accuracies. This improves the utility for all participants. Overall, RealFM is superior at incentivizing contributions compared to state-of-the-art FL mechanisms.

**RealFM Increases Device Utility.** As shown in Figure 5, devices participating in RealFM also boost their utility. The improvement stems from (i) effective accuracy-shaping by RealFM and (ii) the use of non-linear payoff functions $\phi$, which more precisely map the benefit derived from an increase in model accuracy. RealFM's accuracy-shaping function is defined (Theorem 3) to improve device utility if devices contribute more data than they would locally. Incorporating $\phi$ into device utility makes it more realistic and spurs increased data contribution. With utilities that grow increasingly great with improved accuracy, devices are incentivized to contribute more data in order to chase higher model accuracy and subsequent utility.

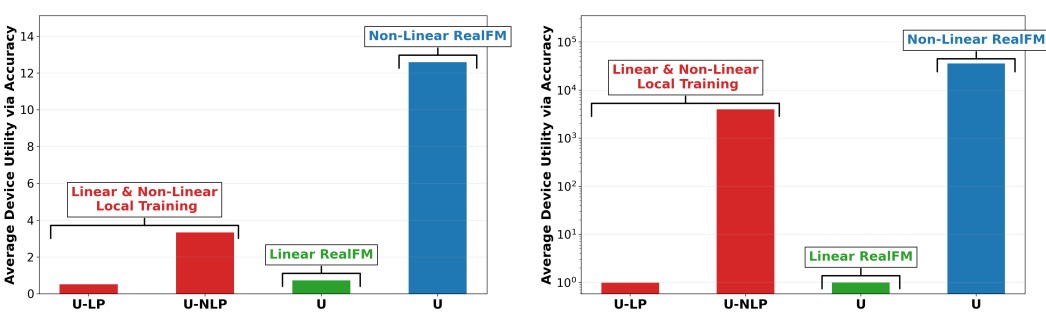

(a) Average Device Utility Gained on CIFAR-10.

(b) Average Device Utility Gained on MNIST.

Figure 5: **Improved Device Utility on Real-World Data.** RealFM improves device utility derived from accuracy by upwards of 6x and 4 *magnitudes* on CIFAR-10 and MNIST respectively. Only uniform experiments are run to save space since the non-uniform results are nearly identical.

## 7 CONCLUSION

Without proper incentives, modern FL frameworks fail to attract devices to participate. Furthermore, current FL mechanisms which incorporate incentives fall victim to the free-rider problem. RealFM is the first FL mechanism which boosts device participation and data contribution all while eliminating the free-rider phenomena without requiring data to be shared. Device and server utility is greatly improved in RealFM by the construction of a novel and realistic utility function which more precisely models the relationship between accuracy and utility. Empirically we found that RealFM's realistic device utility and effective incentive structure, through the use of a novel accuracy-shaping function, resulted in higher-performing models during federated training. As a result, RealFM provides higher utilities to the server and participating devices compared to its peer FL mechanisms.

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

# RealFM Appendix

## A  NOTATION & RELATED WORK

Table 1: Notation Table for RealFM.

| Definition | Notation |
|---|---|
| Number of Devices | $n$ |
| Local FedAvg Training Steps | $h$ |
| Optimal Attainable Accuracy on Learning Task | $a_{opt}$ |
| Difficulty of Learning Task | $k$ |
| Number of Data Points | $m$ |
| Total Data Point Contributions | $\boldsymbol{m}$ |
| Accuracy Function | $a(m)$ |
| Mechanism | $\mathcal{M}$ |
| Server Profit Margin | $p_m$ |
| Server Payoff Function | $\phi_C$ |
| Model Parameters | $\boldsymbol{w}$ |
| Marginal Cost for Device $i$ | $c_i$ |
| Payoff Function for Device $i$ | $\phi_i$ |
| Device $i$ Utility | $u_i$ |
| Local Optimal Device $i$ Utility | $u_i^0$ |
| Rewarded Mechanism Utility for Device $i$ | $u_i^r$ |
| Local Optimal Data Contribution for Device $i$ | $m_i^o$ |
| Mechanism Optimal Data Contribution for Device $i$ | $m_i^*$ |
| Mechanism Model Accuracy Reward | $a^r$ |
| Mechanism Monetary Reward | $R$ |
| Marginal Monetary Reward per Contributed Data Point | $r(\boldsymbol{m})$ |
| Accuracy-Shaping Function for Device $i$ | $\gamma_i$ |

**Federated Mechanisms (Continued).** As detailed in Section 2, there is a wide swath of mechanisms proposed for FL. The works (Zhan et al., 2021; Tu et al., 2022; Zeng et al., 2021; Ali et al., 2023) survey the different methods of incentives present in FL literature. The goal of the presented methods are solely to increase device participation within FL frameworks. The issues of free riding and increased data or gradient contribution are not covered. (Sim et al., 2020; Xu et al., 2021) design model rewards to meet fairness or accuracy objectives. In these works, as detailed below, devices receive models proportionate to the amount of data they contribute but are not incentivized to contribute more data. Our work seeks to incentivize devices to contribute more during training.

**Collaborative Fairness and Federated Shapley Value.** Collaborative fairness in FL (Lyu et al., 2020a;b; Xu & Lyu, 2020; Sim et al., 2020) is closely related to our paper. The works Lyu et al. (2020a;b); Xu & Lyu (2020); Sim et al. (2020) seek to fairly allocate models with varying performance depending upon how much devices contribute to training in FL settings. This is accomplished by determining a "reputation" (a measure of device contributions) for each device, using a hyperbolic sine function, to enforce devices converge to different models relative to their amount of contributions during FL training. Thus, devices who contribute more receive a higher-performing model than those who do not contribute much. There are a few key differences between this line of work and our own, namely: **(1)** Our mechanism incentivizes devices to both participate in training and increase their amount of contributions. There are no such incentives in collaborative fairness. **(2)** We model device and server utility. Unlike collaborative fairness, we do not assume that devices will always participate in FL training. **(3)** Our mechanism design *provably* eliminates the free-rider phenomena unlike collaborative fairness, since devices who try to free-ride receive the same model performance as they would on their own (Equations 12 and 13). **(4)** No monetary rewards are received by devices in current collaborative fairness methods.

While the overarching goal of our work is to incentivize and increase device participation, through mechanism design, within FL, important future research directions remain. Federated Shapley

Value, first proposed in Wang et al. (2020), allows for estimation of the Shapley Value in a FL setting. This is crucial to appraise the data coming from each device and pave the way for rewarding devices with important data during the training process. Data is not equally valuable in real-world settings, and relaxing the assumption of equally valued data within our paper is important future work. We aim to use Wang et al. (2020) to tackle the problem of data heterogeneity.

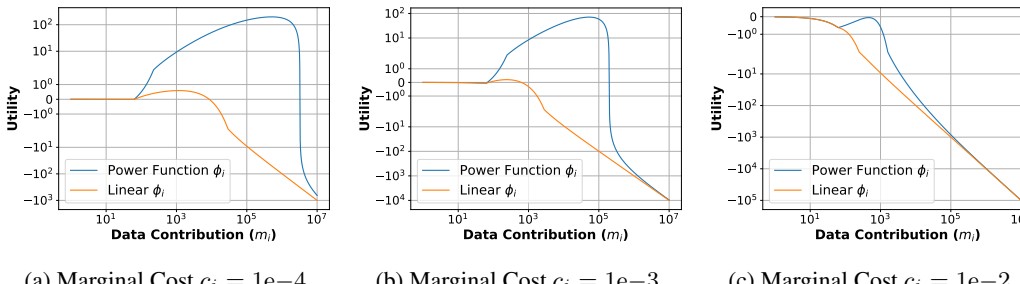

(a) Marginal Cost $c_i = 1e-4$.  (b) Marginal Cost $c_i = 1e-3$.  (c) Marginal Cost $c_i = 1e-2$.

Figure 6: **Utility Functions for Varying Cost and Payoff Functions.** Using both linear, $\phi_i(a) = a$, and power, $\phi_i(a) = \frac{1}{(1-a)^2} - 1$, payoff functions, we compare how device utilities change with rising costs. Once marginal costs $c_i$ become too high, the utility is always negative and devices will not collect data for training. We use $\hat{a}(m)$ as defined in Equation 1, with $a_{opt} = 0.95$ and $k = 1$.

## B  PROOF OF THEOREMS

**Theorem 1.** *Consider a device $i$ with marginal cost per data point $c_i$, accuracy function $\hat{a}(m)$ satisfying Assumption 1, and accuracy payoff $\phi_i$ satisfying Assumption 2. This device will collect the following optimal amount of data $m_i^o$:*

$$m_i^o = \begin{cases} 0 & \text{if } \max_{m_i \geq 0} u_i(m_i) \leq 0, \\ m^*, \text{ such that } \phi_i'(\hat{a}(m^*)) \cdot \hat{a}'(m^*) = c_i & \text{else.} \end{cases} \quad (14)$$

*Proof.* Let $m_0 := \sup\{m \mid \hat{a}(m) = 0\}$ (the point where $a(m)$ begins to increase from 0 and become equivalent to $\hat{a}(m)$). Thus, $\forall m_i > m_0, a(m_i) = \hat{a}(m_i) > 0$. Also, given Assumptions 1 and 2 and Equation 3, $u_i(0) = 0$. The derivative of Equation 3 for device $i$ is,

$$u_i'(m_i) = \phi_i'(a(m_i)) \cdot a'(m_i) - c_i. \quad (15)$$

**Case 1:** $\max_{m_i \geq 0} u_i(m_i) \leq 0$.

Each device $i$ starts with a utility of 0 since by Assumptions 1 and 2 $u_i(0) = 0$. Since $\max_{m_i \geq 0} u_i(m_i) \leq 0$, there is no utility gained by device $i$ to contribute more data. Therefore, the optimal amount of contributions remains at zero: $m_i^* = 0$.

**Case 2:** $\max_{m_i \geq 0} u_i(m_i) > 0$.

*Sub-Case 1: $0 \leq m_i \leq m_0$.* By definition of $m_0$, $a(m_i) = 0 \ \forall m_i \in [0, m_0]$. Therefore, from Equation 3 and Assumptions 1 and 2, $u_i(m_i) = -c_i m_i < 0 \ \forall m_i \in (0, m_0]$. Since $u_i(0) = 0$ and $u_i(m_i) < 0$ for $m_i > 0$, device $i$ will not collect any contribution: $m_i^* = 0$.

*Sub-Case 2: $m_i > m_0$.* Since $\forall m_i > m_0, a(m_i) = \hat{a}(m_i) > 0$, Equation 15 becomes,

$$u_i'(m_i) = \phi_i'(\hat{a}(m_i)) \cdot \hat{a}'(m_i) - c_i. \quad (16)$$

We begin by showing that $\phi_i(\hat{a}(m_i))$ is bounded. By Assumption 1, $\hat{a}(m_i) < a_{opt} < 1 \ \forall m_i$. Thus, $\phi_i(\hat{a}(m_i)) < \phi_i(a_{opt}) < \infty \ \forall m_i$ since $\hat{a}(m_i)$ and $\phi_i$ are non-decreasing and continuous by Assumptions 1 and 2. Due to $\phi_i(\hat{a}(m_i))$ being concave, non-decreasing, and bounded (by Assumption 2), we have from Equation 16 that the limit of its derivative,

$$\lim_{m_i \to \infty} \phi_i'(\hat{a}(m_i)) \cdot \hat{a}'(m_i) = 0 \implies \lim_{m_i \to \infty} u_i'(m_i) = -c_i < 0. \quad (17)$$

Since $\phi_i(\hat{a}(m_i))$ is concave and non-decreasing, its gradient $\phi_i'(\hat{a}(m_i)) \cdot \hat{a}'(m_i)$ is maximized when $m_i = m_0$ and is non-increasing afterwards. Using the maximal derivative location $m_i = m_0$ in union with Case 2 ($\max_{m_i \geq 0} u_i(m_i) > 0$) and Sub-Case 2 ($u_i(m) \leq 0 \ \forall m \in [0, m_0]$) yields $u_i'(m_0) > 0$ (the derivative must be positive in order to increase utility above 0).

Now that $u_i'(m_0) > 0$, $\lim_{m_i \to \infty} u_i'(m_i) < 0$, and $\phi_i'(\hat{a}(m_i)) \cdot \hat{a}'(m_i)$ is non-increasing, there must exist a maximum $m_i = m^*$ such that $\phi_i'(\hat{a}(m^*)) \cdot \hat{a}'(m^*) = c_i$. $\qquad\square$

> **Corollary 2.** *Under uniform payoff functions $\phi_i(a(m)) = \phi(a(m)) \ \forall i$, two devices $j, k$ with respective marginal costs $c_j \leq c_k$ satisfy $u_j(m_j^o) \geq u_k(m_k^*)$ and $m_j^* \geq m_k^*$.*

Corollary 2 states that a device with lower marginal cost has higher utility and would contribute more data than a device with a higher marginal cost.

*Proof.* With uniform payoff functions, each device $i$'s utility and utility derivative become

$$u_i(m_i) = \phi(a(m_i)) - c_i m_i, \quad u_i'(m_i) = \phi'(a(m_i)) \cdot a'(m_i) - c_i. \tag{18}$$

Due to $\phi_i(a(m))$ being concave and non-decreasing, its derivative $\phi'(a(m_i)) \cdot a'(m_i)$ is non-negative and non-increasing. Let $m_k^o$ be the optimal amount of data contribution by device $k$.

**Case 1:** $c_j = c_k$. By Equation 18, if $c_j = c_k$ then $0 = u_k'(m_k^o) = u_j'(m_k^o)$. This implies $m_k^o = m_j^o$ and subsequently $u_k(m_k^o) = u_j(m_k^o)$.

**Case 2:** $c_j < c_k$. By Equation 18, if $c_j < c_k$, then $0 = u_k'(m_k^o) < u_j'(m_k^o)$. Since $\phi(a(m_i))$ is concave and non-decreasing, its derivative is non-increasing with its limit going to 0. Therefore, more data $\epsilon > 0$ must be collected in order for $u_j'$ to reach zero (*i.e.* $u_j'(m_k^o + \epsilon) = 0$). This implies that $m_j^o = m_k^o + \epsilon > m_k^o$. Furthermore, since $0 = u_k'(m_k^o) < u_j'(m_k^o)$, the utility for device $j$ is still increasing at $m_k^o$ and is fully maximized at $m_j^o$. This implies that $u_j(m_j^o) > u_k(m_k^o)$. $\qquad\square$

**Theorem 2.** *Consider a feasible mechanism $\mathcal{M}$ returning utility $[\mathcal{M}^U(m_i; \boldsymbol{m}_{-i})]_i$ to device $i$ ($[\mathcal{M}^U(0; \boldsymbol{m}_{-i})]_i = 0$). Define the utility of a participating device $i$ as,*

$$u_i^r(m_i; \boldsymbol{m}_{-i}) := [\mathcal{M}^U(m_i; \boldsymbol{m}_{-i})]_i - c_i m_i. \tag{19}$$

*If $u_i^r(m_i, \boldsymbol{m}_{-i})$, is quasi-concave for $m_i \geq m_i^u := \inf\{m_i \mid [\mathcal{M}^U(m_i; \boldsymbol{m}_{-i})]_i > 0\}$ and continuous in $\boldsymbol{m}_{-i}$, then a pure Nash equilibrium with $\boldsymbol{m}^{eq}$ data contributions exists such that,*

$$u_i^r(\boldsymbol{m}^{eq}) = [\mathcal{M}^U(\boldsymbol{m}^{eq})]_i - c_i \boldsymbol{m}_i^{eq} \geq u_i^r(m_i; \boldsymbol{m}_{-i}^{eq}) = [\mathcal{M}^U(m_i; \boldsymbol{m}_{-i}^{eq})]_i - c_i m_i \text{ for } m_i \geq 0. \tag{20}$$

*Proof.* We start the proof by examining two scenarios.

**Case 1:** $\max_{m_i} u_i^r(m_i; \boldsymbol{m}_{-i}) \leq 0$.

In the case where the marginal cost of producing updates $-c_i m_i$ is so large that the device utility $u_i$ will always be non-positive, the best response $[B(\boldsymbol{m})]_i$ given a set of contributions $\boldsymbol{m}$ for device $i$ is,

$$[B(\boldsymbol{m})]_i = \arg\max_{m_i \geq 0} u_i^r(m_i; \boldsymbol{m}_{-i}) = 0. \tag{21}$$

As expected, device $i$ will not perform any updates $\boldsymbol{m}_i^{eq} = 0$. Therefore, Equation 8 is fulfilled since as $\max_{m_i} u_i^r(m_i; \boldsymbol{m}_{-i}) \leq 0$ we see,

$$[\mathcal{M}^U(\boldsymbol{m}^{eq})]_i - c_i \boldsymbol{m}_i^{eq} = [\mathcal{M}^U(0; \boldsymbol{m}_{-i})]_i - c_i(0) = 0 \geq [\mathcal{M}^U(m_i; \boldsymbol{m}_{-i}^{eq})]_i - c_i m_i \text{ for all } m_i \geq 0. \tag{22}$$

**Case 2:** $\max_{m_i} u_i(m_i; \boldsymbol{m}_{-i}) > 0$.

Denote $m_i^u := \inf\{m_i \mid [\mathcal{M}^U(m_i; \boldsymbol{m}_{-i})]_i > 0\}$. On the interval of integers $m_i \in [0, m_i^u]$, device $i$'s utility is non-positive,

$$u_i^r(m_i; \boldsymbol{m}_{-i}) = [\mathcal{M}^U(m_i; \boldsymbol{m}_{-i})]_i - c_i m_i = -c_i m_i \leq 0, \quad \forall m_i \in [0, m_i^u]. \tag{23}$$

For $m_i \geq m_i^u$, we have that $u_i(m_i; \boldsymbol{m})$ is quasi-concave. Let the best response for a given set of contributions $\boldsymbol{m}_{-i}$ for device $i$ be formally defined as,

$$[B(\boldsymbol{m})]_i := \arg\max_{m_i \geq 0} u_i^r(m_i; \boldsymbol{m}_{-i}) = \arg\max_{m_i \geq 0} [\mathcal{M}^U(m_i; \boldsymbol{m}_{-i})]_i - c_i m_i. \tag{24}$$

Suppose there exists a fixed point $\tilde{\boldsymbol{m}}$ to the best response, $\tilde{\boldsymbol{m}} \in B(\tilde{\boldsymbol{m}})$. This would mean that $\tilde{\boldsymbol{m}}$ is an equilibrium since by Equation 24 we have for any $m_i \geq 0$,

$$[\mathcal{M}^U(\tilde{m}_i; \tilde{\boldsymbol{m}}_{-i})]_i - c_i \tilde{m}_i \geq [\mathcal{M}^U(m_i; \boldsymbol{m}_{-i})]_i - c_i m_i. \tag{25}$$

Thus, now we must show that $B$ has a fixed point (which is subsequently an equilibrium). To do so, we first determine a convex and compact search space. As detailed in Case 1, $u_i^r(0, \boldsymbol{m}_{-i}) = 0$. Therefore, we can bound $0 \leq \max_{m_i} u_i^r(m_i, \boldsymbol{m}_{-i})$. Since $\mathcal{M}$ is feasible, $\mathcal{M}(\boldsymbol{m})$ is bounded above by $\mathcal{M}_{max}^U$. Thus, we find

$$0 \leq \max_{m_i} u_i^r(m_i; \boldsymbol{m}_{-i}) \leq \mathcal{M}_{max}^U - c_i m_i. \tag{26}$$

Rearranging yields $m_i \leq \mathcal{M}_{max}^U/c_i$. Since $m_i \geq m_i^u$, we can restrict our search space to $\mathcal{C} := \prod_j [m_j^u, \mathcal{M}_{max}^U/c_j] \subset \mathbb{R}^n$, where our best response mapping is now over $B : \mathcal{C} \to 2^{\mathcal{C}}$.

**Lemma 1** (Kakutani's Theorem). *Consider a multi-valued function $F : \mathcal{C} \to 2^{\mathcal{C}}$ over convex and compact domain $\mathcal{C}$ for which the output set $F(\boldsymbol{m})$ (i) is convex and closed for any fixed $\boldsymbol{m}$, and (ii) changes continuously as we change $\boldsymbol{m}$. For any such $F$, there exists a fixed point $\boldsymbol{m}$ such that $\boldsymbol{m} \in F(\boldsymbol{m})$.*

Since within this interval of $m_i$ $u_i^r(m_i, \boldsymbol{m}_{-i})$ is quasi-concave, $B(\boldsymbol{m})$ must be continuous in $\boldsymbol{m}$ (from Acemoglu & Ozdaglar (2009)). Now by applying Lemma 1, Kakutani's fixed point theorem, there exists such a fixed point $\tilde{\boldsymbol{m}}$ such that $\tilde{\boldsymbol{m}} \in B(\tilde{\boldsymbol{m}})$ where $\tilde{m}_i \geq m_i^u$. Since $\max_{m_i} u_i^r(m_i; \boldsymbol{m}_{-i}) > 0$ and $u_i^r(m_i; \boldsymbol{m}_{-i}) \leq 0$ for $m_i \in [0, m_i^u]$, $\tilde{m}_i$ is certain to not fall within $[0, m_i^u] \, \forall i$ due to the nature of the arg max in Equation 24. Therefore, Equation 25 holds as the fixed point $\tilde{\boldsymbol{m}}$ exists and is the equilibrium of $\mathcal{M}$. $\qquad\square$

**Theorem 3.** *Consider a device $i$ with marginal cost $c_i$ and accuracy payoff function $\phi_i$ satisfying Assumption 2. Denote device $i$'s optimal local data contribution as $m_i^o$ and its subsequent accuracy $\bar{a} = a(m_i^o)$. Define the derivative of $\phi_i(a)$ with respect to $a$ as $\phi_i'(a)$. For any $\epsilon \to 0^+$ and marginal server reward $r(\boldsymbol{m}) \geq 0$, device $i$ has the following accuracy-shaping function $\gamma_i(m)$ for $m \geq m_i^o$,*

$$\gamma_i := \begin{cases} \frac{-\phi_i'(\bar{a}) + \sqrt{\phi_i'(\bar{a})^2 + 2\phi_i''(\bar{a})(c_i - r(\boldsymbol{m}) + \epsilon)(m - m_i^o)}}{\phi_i''(\bar{a})} & \text{if } \phi_i''(\bar{a}) > 0 \\ \frac{(c_i - r(\boldsymbol{m}) + \epsilon)(m - m_i^o)}{\phi_i'(\bar{a})} & \text{if } \phi_i''(\bar{a}) = 0 \end{cases} \tag{27}$$

*Given the defined $\gamma_i(m)$, the following inequality is satisfied for $m \in [m_i^o, m_i^*]$,*

$$\phi_i(a(m_i^o) + \gamma_i(m)) - \phi_i(a(m_i^o)) > (c_i - r(\boldsymbol{m}))(m - m_i^o). \tag{28}$$

*The new optimal contribution is $m_i^* := \{m \geq m_i^o \mid a(m + \sum_{j \neq i} m_j) = a(m_i^o) + \gamma_i(m)\}$. Accuracy shaping improves device $i$'s data contribution $m_i^* \geq m_i^o$ for any contribution $\boldsymbol{m}_{-i}$.*

*Proof.* By the mean value version of Taylor's theorem we have,

$$\phi_i(\bar{a} + \gamma_i) = \phi_i(\bar{a}) + \gamma_i \phi_i'(\bar{a}) + 1/2 \gamma_i^2 \phi_i''(z), \quad \text{for some } z \in [\bar{a}, \bar{a} + \gamma_i]. \tag{29}$$

Since $\phi_i(a)$ is both increasing and convex with respect to $a$,

$$\phi_i(\bar{a} + \gamma_i) - \phi_i(\bar{a}) \geq \gamma_i \phi_i'(\bar{a}) + 1/2 \gamma_i^2 \phi_i''(\bar{a}). \tag{30}$$

In order to ensure $\phi_i(\bar{a} + \gamma) - \phi_i(\bar{a}) > (c_i - r(\boldsymbol{m}))(m - m_i^o)$, we must select $\gamma$ such that,

$$\gamma_i \phi_i'(\bar{a}) + 1/2 \gamma_i^2 \phi_i''(\bar{a}) > (c_i - r(\boldsymbol{m}))(m - m_i^o). \tag{31}$$

**Case 1:** $\phi_i''(\bar{a}) = 0$. In this case, Equation 31 becomes,

$$\gamma_i \phi_i'(\bar{a}) > (c_i - r(\boldsymbol{m}))(m - m_i^o). \tag{32}$$

In order for Equation 31, and thereby Equation 11, to hold we select $\epsilon \to 0^+$ such that,

$$\gamma_i := \frac{(c_i - r(\boldsymbol{m}) + \epsilon)(m - m_i^o)}{\phi_i'(\bar{a})}. \tag{33}$$

**Case 2:** $\phi_i''(\bar{a}) > 0$. Determining when the left- and right-hand sides of Equation 31 are equal is equivalent to solving the quadratic equation for $\gamma_i$,

$$\gamma_i = \frac{-\phi_i'(\bar{a}) \pm \sqrt{\phi_i'(\bar{a})^2 + 2\phi_i''(\bar{a})(c_i - r(\boldsymbol{m}))(m - m_i^o)}}{\phi_i''(\bar{a})} \tag{34}$$

$$= \frac{-\phi_i'(\bar{a}) + \sqrt{\phi_i'(\bar{a})^2 + 2\phi_i''(\bar{a})(c_i - r(\boldsymbol{m}))(m - m_i^o)}}{\phi_i''(\bar{a})}. \tag{35}$$

The second equality follows from $\gamma_i$ having to be positive. In order for Equation 31, and thereby Equation 11, to hold we select $\epsilon \to 0^+$ such that,

$$\gamma_i := \frac{-\phi_i'(\bar{a}) + \sqrt{\phi_i'(\bar{a})^2 + 2\phi_i''(\bar{a})(c_i - r(\boldsymbol{m}) + \epsilon)(m - m_i^o)}}{\phi_i''(\bar{a})} \tag{36}$$

As a quick note, for $m = m_i^o$ one can immediately see that $\gamma_i(m) = 0$. To finish the proof, now that Equation 11 is proven to hold for the prescribed $\gamma_i$, device $i$ is incentivized to contribute more as the added utility $\phi_i(\bar{a} + \gamma_i) - \phi_i(\bar{a})$ is larger than the incurred cost $(c_i - r(\boldsymbol{m}))(m - m_i^o)$. There is a limit to this incentive, however. The maximum value that $\gamma_i$ can be is bounded by the accuracy from all contributions: $a(m_i^o) + \gamma_i \leq a(\sum_j m_j)$. Thus, device $i$ reaches a new optimal contribution $m_i^*$ which is determined by,

$$m_i^* := \{m \geq m_i^o \mid a(m + \sum_{j \neq i} m_j) = a(m_i^o) + \gamma_i(m)\} \geq m_i^o. \tag{37}$$

$\square$

**Theorem 4.** *Our realistic federated mechanism $\mathcal{M}$, defined in Equation 13, performs accuracy-shaping with $\gamma_i$ defined in Theorem 3 for each device $i \in [n]$ and some small $\epsilon \to 0^+$. As such, $\mathcal{M}$ is Individually Rational (IR) and has a unique Nash equilibrium at which device $i$ will contribute $m_i^* \geq m_i^o$ updates, thereby eliminating the free-rider phenomena. Furthermore, since $\mathcal{M}$ is IR, devices are incentivized to participate as they gain equal to or more utility than by not participating.*

*Proof.* We first prove existence of a unique Nash equilibrium by showcasing how our mechanism $\mathcal{M}$ fulfills the criteria laid out in Theorem 2. The criteria in Theorem 2 largely surrounds the utility of a participating device $i$,

$$u_i^r(m_i; \boldsymbol{m}_{-i}) := [\mathcal{M}^U(m_i; \boldsymbol{m}_{-i})]_i - c_i m_i. \tag{38}$$

**Feasibility.** Before beginning, we note that $\mathcal{M}$ trivially satisfies the only non-utility requirement that $[\mathcal{M}^U(0; \boldsymbol{m}_{-i})]_i = 0$ (as $a(0) = \phi_i(0) = 0$). As shown in Equation 13, $\mathcal{M}$ returns accuracies between 0 and $a(\sum \boldsymbol{m})$ to all devices. This satisfies the bounded accuracy requirement. Furthermore, the utility provided by our mechanism $\mathcal{M}^U$ is bounded as well. Since $a_{opt}$ is the largest attained accuracy by our defined accuracy function $\hat{a}(m)$ and $a_{opt} < 1$, the maximum utility is $\phi_i(a_{opt}) < \infty$. Now, that $\mathcal{M}$ is proven to be Feasible, we only need the following to prove that $\mathcal{M}$ has a pure equilibrium: **(1)** $u_i^r(m_i; \boldsymbol{m}_{-i})$ is continuous in $\boldsymbol{m}_{-i}$ and **(2)** quasi-concave for $m_i \geq m_i^u := \inf\{m_i \mid [\mathcal{M}^U(m_i; \boldsymbol{m}_{-i})]_i > 0\}$.

**Continuity.** By definition of $u_i^r(m_i; \boldsymbol{m}_{-i})$, Equation 38, we only need to consider $[\mathcal{M}^U(m_i; \boldsymbol{m}_{-i})]_i$ since that is the only portion affected by $\boldsymbol{m}_{-i}$. By definition of the utility returned by our mechanism $\mathcal{M}$, shown in Equation 12, no discontinuities arise for a fixed $m_i$ and varying $\boldsymbol{m}_{-i}$. By assumptions on continuity in Assumptions 1 and 2, $\phi_i(a(m))$ is continuous for all $m$. Thus, for non-zero utility (zero utility would lead to zero reward), we find the marginal monetary reward function $r(\boldsymbol{m})$ in Equation 9 is continuous. Therefore, each piecewise component of $[\mathcal{M}^U(m_i; \boldsymbol{m}_{-i})]_i$ is continuous since they are sums of continuous functions. Finally, we show that the piecewise functions connect with each other continuously. The accuracy-shaping function $\gamma_i$ is defined such that $\gamma_i(m_i^o) = 0$ and $a(m_i^o) + \gamma_i(m_i^*) = a(\sum \boldsymbol{m})$, which finishes proof of continuity.

**Quasi-Concavity.** For all values of $m_i \geq m_i^u$, our mechanism $\mathcal{M}$ produces positive utility. By construction, our mechanism $\mathcal{M}$ is strictly increasing for $m_i \geq m_i^u$. Our mechanism $\mathcal{M}$ returns varying utilities within three separate intervals. While piece-wise, these intervals are continuous and $\mathcal{M}$ is strictly increasing with respect to $m_i$ in each. The first interval, consisting of the concave function $\phi_i(a(m_i))$, is quasi-concave by construction. The second interval consists of a linear function $r(\boldsymbol{m}) \cdot (m_i - m_i^o)$ added to a quasi-concave $\phi_i(a(m_i) + \gamma_i(m_i))$ function, resulting in a quasi-concave function (note that $\phi_i(\hat{a}(m_i))$ is concave). Finally, the third interval consists of a linear function $r(\boldsymbol{m}) \cdot (m_i - m_i^*)$ added to a concave function $\phi_i(a(m_i))$, which is also quasi-concave. In sum, this makes $[\mathcal{M}^U(m_i; \boldsymbol{m}_{-i})]_i$ a quasi-concave function. Since $-c_i m_i$ is a linear function, the utility of a participating device $u_i^r(m_i; \boldsymbol{m}_{-i})$ will also be quasi-concave function, as the sum of a linear and quasi-concave function is quasi-concave.

**Existence of Pure Equilibrium with Increased Data Contribution.** Since $[\mathcal{M}^U(\boldsymbol{m})]_i$ satisfies feasibility, continuity, and quasi-concavity requirements, $\mathcal{M}$ is guaranteed to have a pure Nash equilibrium by Theorem 2. Furthermore, since $\mathcal{M}$ performs accuracy-shaping with $\gamma_i$ prescribed in Theorem 2, it is guaranteed that each device $i$ will produce $m_i^* \geq m_i^o$ updates.

**Individually Rational (IR).** We prove $\mathcal{M}$ is IR by looking at each piecewise portion of Equation 13:

*Case 1: $m_i \leq m_i^o$ (Free-Riding).* When $m_i \leq m_i^o$, a device would attempt to provide as much or less than the amount of contribution which is locally optimal. The hope for such strategy would be free-riding: enjoy the performance of a well-trained model as a result of federated training while providing few (or zero) data points in order to save costs. Our mechanism avoids the free rider problem trivially by returning a model with an accuracy that is proportional to the amount of data contributed by the device. This is shown in Equation 13, as devices receive a model with accuracy $a(m_i)$ if $m_i \leq m_i^o$ (*i.e.*, devices are rewarded with a model equivalent to one that they could've trained themselves if they fail to contribute an adequate amount of data). In this case, devices receive the same model accuracy as they would've on their own and thus IR is satisfied in this case.

*Case 2: $m_i \in (m_i^o, m_i^*]$.* Via the results of Theorem 3, the accuracy of the model returned by $\mathcal{M}$ when $m_i \in (m_i^o, m_i^*]$ is greater than a model trained by device $i$ on $m_i$ local contributions. Mathematically this is described as $a^r(m_i) = a(m_i) + \gamma_i(m_i) > a(m_i)$ for $m_i \in (m_i^o, m_i^*]$. Since $\phi_i$ is increasing, this ensures that Equation 6 holds.

*Case 3: $m_i \geq m_i^*$.* By Theorem 3, by definition of $m_i^*$ when $m_i = m_i^*$ then the accuracy of a returned model by $\mathcal{M}$ is equal to $a(\sum_j m_j)$. Therefore, given a fixed set of contributions from all other devices $\boldsymbol{m}_{-i}$, device $i$ will still attain a model with accuracy $a(\sum_j m_j)$ for $m_i \geq m_i^*$ (since the limits of accuracy shaping have been reached for the given contributions). Due to this, Equation 6 trivially holds as $a(\sum_j m_j) \geq a(m_i)$. □

## B.1 ACCURACY MODELING

Our model for accuracy stems from Example 2.1 in Karimireddy et al. (2022), which in turn comes from Theorem 11.8 in Mohri et al. (2018). Below is the mentioned generalization bound,

**Proposition 1** (Generalization Bounds, Karimireddy et al. (2022) Example 2.1). *Suppose we want to learn a model $h$ from a hypothesis class $\mathcal{H}$ which minimizes the error over data distribution $\mathcal{D}$, defined to be $R(h) := \mathbb{E}_{(x,y) \sim \mathcal{D}}[e(h(x), y)]$, for some error function $e(\cdot) \in [0, 1]$. Let such an optimal model have error $(1 - a_{opt}) \leq 1$. Now, given access to $\{(x_l, y_l)\} l \in [m]$ which are $m$ i.i.d. samples from $\mathcal{D}$, we can compute the empirical risk minimizer (ERM) as $\hat{h}_m = \arg\min_{h \in \mathcal{H}} \sum_{l \in [m]} e(h(x), y)$. Finally, let $k > 0$ be the pseudo-dimension of the set of functions $\{(x, y) \to e(h(x), y) : h \in \mathcal{H}\}$, which is a measure of the difficulty of the learning task. Then, standard generalization bounds imply that with probability at least 99% over the sampling of the data, the accuracy is at least*

$$1 - R(\hat{h}_m) \geq \left\{ \hat{a}(m) := a_{opt} - \frac{\sqrt{2k(2 + \log(m/k))} + 4}{\sqrt{m}} \right\}. \tag{39}$$

*A simplified expression for our analytic analysis use is,*

$$\hat{a}(m) = a_{opt} - 2\sqrt{k/m}. \tag{40}$$

## C  EXPERIMENTAL RESULTS CONTINUED

In this section we provide durther details into how our image classification experiments were run. As a note, we ran each experiment three times, varying the random seeds. All bar plots in our paper showcase the mean results of the three experiments. In Figures 7 and 8, we plot error bars. The error bars are quite thin, as the results did not vary much between each of the three experiments.

### C.1  ADDITIONAL EXPERIMENTAL DETAILS

The experimental process involved careful tuning of our theoretical accuracy function $\hat{a}(m)$ in order to match the empirical accuracy results we found. In fact, for CIFAR-10 we use $\hat{a}(m)$ defined in Equation 1 with carefully selected values for $k$ and $a_{opt}$ to precisely reflect the empirical results found for our CIFAR-10 training. For MNIST, however, an accuracy function of $\hat{a}(m) = a_{opt} - 2\sqrt{k/m}$ was more reflective of the ease of training on MNIST. Overall, to ensure precise empirical results, we followed the following process for each experiment (for both CIFAR-10 and MNIST as well as for 8 and 16 devices):

1. Determine $a_{opt}$ and $k$ such that $\hat{a}(m)$ is tuned to most precisely reflect our empirical results.
2. Select an amount of data that each device should have *in expectation of varying costs and payoffs* (this amount should ensure that there is enough unique datapoints to share amongst devices).
3. From the prescribed amount of data and $a(m)$, derive the marginal cost $c_e$ required to produce the desired data quantity.
4. For non-uniform experiments, draw a marginal cost at random from a Gaussian with mean $c_e$ and/or draw a payoff function $\phi_i$ with a $z_i$ sampled within $[0.9, 1.1]$ (detailed in Section 6).
5. Save an initial model for training.
6. Train this initial model locally on each device until convergence, generating $a_{local}$.
7. Using the initial model, train the model in a federated manner until convergence, generating $a_{fed}$.
8. Using the accuracy-shaping function, compute the amount of additional data contributions required to raise $a_{local}$ to $a_{fed}$. *This is the incentivized amount of added contributions.*

We also use the expected payoff function for the central server: $\phi_C = 1/(1-a)^2 - 1$. Within experiments, for simplicity, we set the profit margin $p_m = 1$ (greedy server). Below, we detail the hyper-parameters used in our CIFAR-10 and MNIST experiments.

Table 2: Hyper-parameters for CIFAR-10 Experiments.

| Model | Batch Size | Learning Rate | $a_{opt}$ | $k$ | Epochs | Local FedAvg Steps $h$ |
|---|---|---|---|---|---|---|
| ResNet18 | 128 | 1e-3 | 0.9 | 18 | 100 | 6 |

Table 3: Hyper-parameters for MNIST Experiments.

| Model | Batch Size | Learning Rate | $a_{opt}$ | $k$ | Epochs | Local FedAvg Steps $h$ |
|---|---|---|---|---|---|---|
| CNN | 128 | 1e-3 | 0.995 | 0.25 | 50 | 6 |

### C.2  ADDITIONAL EXPERIMENTAL RESULTS

It is interesting to note how well RealFM performs on MNIST (in terms of the vastly improved utility seen in Figures 4 and 10) while FedAvg only improves model accuracy by a mere couple of percentage points. The reason stems from the payoff function $\phi_i$ which heavily rewards models that have accuracies close to 100%. This scenario is rational in real-world settings. Competing companies in industry will all likely have models which are high-performing (above 95% accuracy). Since the competition is stiff, companies with the best model performance will likely attract the most customers since their product is the best. Therefore, the utility for achieving model performance close to 100% should become larger and larger as one gets closer to 100%.

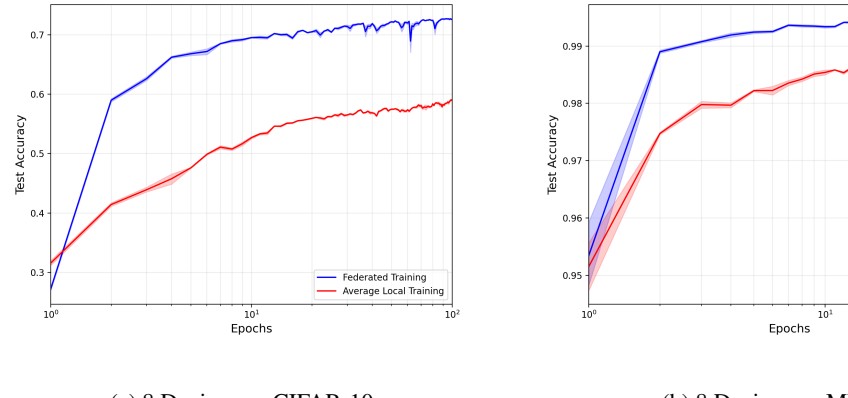

(a) 8 Devices on CIFAR-10.          (b) 8 Devices on MNIST.

Figure 8: **Test Accuracy for Local versus Federated Training on CIFAR-10 and MNIST.** As expected, FedAvg greatly outperforms Local Training by nearly 14% for 8 devices respectively on CIFAR-10. MNIST is an easier dataset to classify, so both Local Training and FedAvg do great jobs at classifying digits. However, FedAvg does perform at a 99+% accuracy whereas Local Training sits around 98%. While this may seem to be a small gap, the utility gained by having a nearly perfect classification scheme is much greater than that having only 98%.

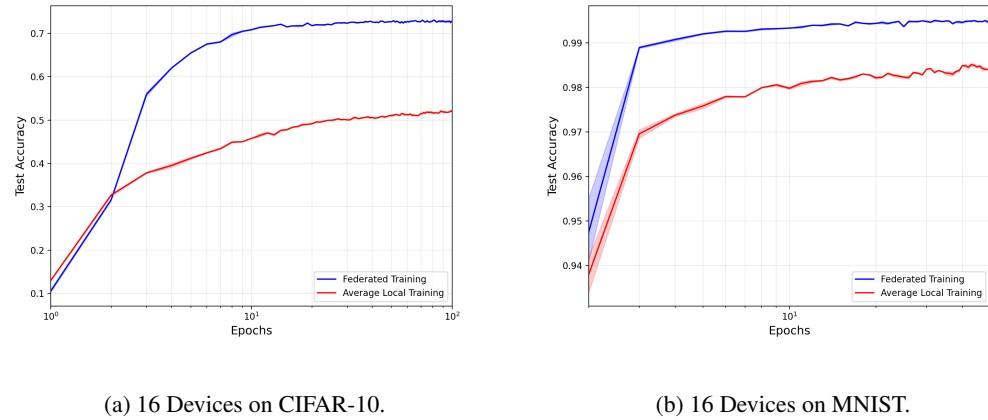

(a) 16 Devices on CIFAR-10.          (b) 16 Devices on MNIST.

Figure 7: **Test Accuracy for Local versus Federated Training on CIFAR-10 and MNIST.** As expected, FedAvg greatly outperforms Local Training by nearly 20% for 16 devices respectively on CIFAR-10. MNIST is an easier dataset to classify, so both Local Training and FedAvg do great jobs at classifying digits. However, FedAvg does perform at a 99+% accuracy whereas Local Training sits around 98%. While this may seem to be a small gap, the utility gained by having a nearly perfect classification scheme is much greater than that having only 98%.

**8 Device Image Classification Results.** Below we provide CIFAR-10 and MNIST results for 8 devices. These plots mirror those shown in Section 6 for 16 devices. In both cases, we find that utility sharply improves for the central server and participating devices. Data contribution also improves for CIFAR-10 and MNIST. Under all scenarios RealFM performs the best compared to all other baselines. First, we start with the test accuracy curves for both datasets.

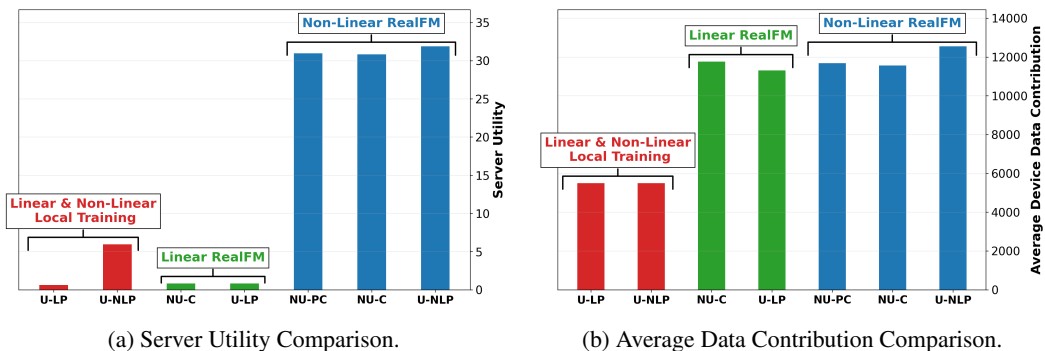

(a) Server Utility Comparison.

(b) Average Data Contribution Comparison.

Figure 9: **Server Utility for** 8 **Devices on CIFAR-10.**

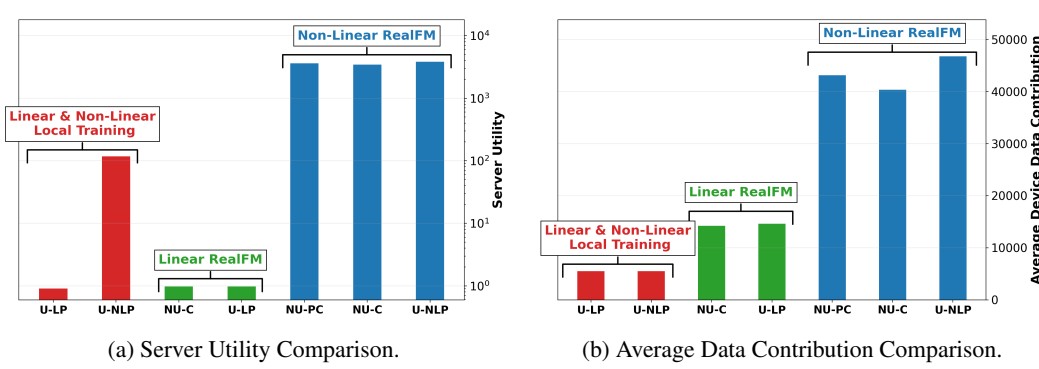

(a) Server Utility Comparison.

(b) Average Data Contribution Comparison.

Figure 10: **Server Utility for** 8 **Devices on MNIST.**

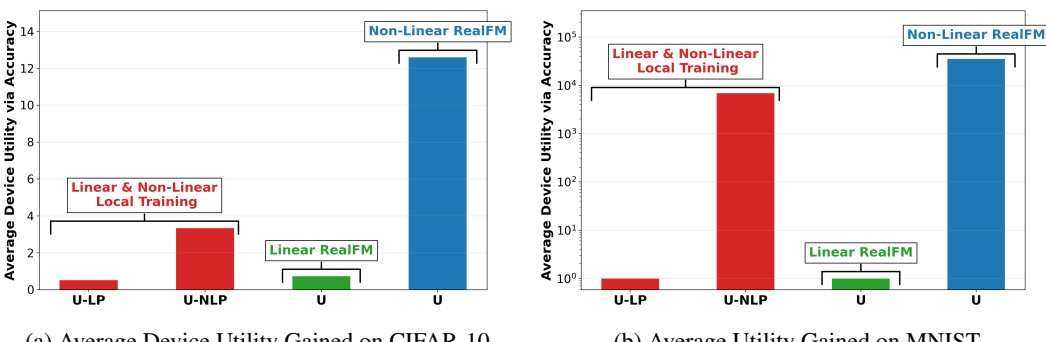

(a) Average Device Utility Gained on CIFAR-10.

(b) Average Utility Gained on MNIST.

Figure 11: **Improved Device Utility for** 8 **Devices on Real-World Data.**

