# OpenReview forum: "RealFM: A Realistic Mechanism to Incentivize Data Contribution and Device Participation"
_ICLR.cc/2024/Conference — Submitted to ICLR 2024_

### Official Review · Reviewer_gdSK · 2023-10-29

**Soundness:** 3 good
**Presentation:** 3 good
**Contribution:** 2 fair
**Rating:** 5
**Confidence:** 4

**Summary:**

This work builds on [Karimireddy et.al. (2022)] and develops a federated mechanism that incentivizes truthful participation and data contribution while provably removing the free-rider problem. The methods eliminate the requirement for data sharing while achieving a high-quality global model following the developed reward protocol - per non-linear modelling of utility with accuracy- that compensates rational devices for participation with more data.

**Strengths:**

This work develops a federated mechanism that incentivizes truthful participation and data contribution while provably removing the free-rider problem. The methods eliminate the requirement for data sharing while achieving a high-quality global model following the developed reward protocol - per non-linear modelling of utility with accuracy- that compensates rational devices for participation with more data.

**Weaknesses:**

1. This work builds on  [Karimireddy et.al. (2022)]. The derivation to model accuracy in (1) and (2), following [Karimireddy et al., 2022] and moreover, [Mohri et al., 2018] assumes for m i.i.d. samples. Will it enjoy generalization to the non-i.i.d. case?
2. Some inconsistencies and incompleteness due to unclear representation. For instance, the authors mention generalizing $\phi_i$ to become a convex and increasing function (but in which variable), and also, in actuality, the utility is modelled as a concave function? Following that, Assumption 2  needs further clarity. Another concern is Fig. 2, which is not well-justified for the said "payoff function. I don't see this claim has been experimentally validated apart from numerical evaluation.
3. How a_\textrm{opt} is derived/obtained?
4. If the profit margin of the central server, defined as p_m, is fixed and known by all devices, I am not sure why it is used after all. (in the later experiments, this is used to indicate the degree to which the server is greedy). Assuming this is unknown would be an interesting analysis. Also, the definition of profit margin should be made more rigorous.
5. As the authors mentioned, it is hard to bond the composition function $\phi$, and I am still not sure how, particularly with this framework, we can ensure increased data production leads to better payoffs. In principle, one has to factor in the quality of data or limit combinatorial properties.
6. Evaluations:
 - Indicate the choice of parameters: z_i.
- Can the authors be more precise on the whole experimental setup? Specifically, how "more data" is used in practice? Did I miss something?
 - In its current form, the discussion on device utility is not rigorous, for instance, how it is evaluated. It would be interesting to evaluate per-device performance, apart from the server's utility, as Top-1 accuracy with optimal data contribution and, later, under the influence of accuracy shaping (with RealFM?). Then, can we get the best of both worlds, a high-quality model with improved generalization/personalization performance? Please comment.

Minor suggestion:
In the Fig. 1 caption, it should be said, in my understanding, as such that RealFM ensures better utility for a "truthful" participation instead.

**Questions:**

Please see the questions posed in the weakness section.

---

> ### Author Response · Authors · 2023-11-16
> **Reviewer gdSK Rebuttal**
>
> We thank reviewer gdSK for their thoughtful review and look forward to our discussion. We begin by reaffirming the novelty of our work before addressing the questions raised.
>
> ### Novelty of Contributions
>
>  **Our work is the first FL mechanism which more realistically models device utility (non-linear function of model accuracy) and provably improves device participation and contributions (eliminating the free-rider effect) without the need of a payment mechanism prior to training**. We sought to design a foundational FL mechanism which improves upon previous FL mechanisms by:
> 1. Modeling device utility in a more realistic manner which enables flexible modeling of diverse device utilities that accommodate non-linear relationships between model accuracy and utility.
>      - Allowing this non-linear relationship within device utility ***requires major changes*** to the accuracy-shaping procedure, the process in which the server provides devices increased model accuracy in exchange for an increase in contributions, at the heart of the incentive mechanism (Theorem 4).
>      - The altered accuracy-shaping function and its provable guarantees in Theorem 3 ***are novel***.
>      - The accuracy-shaping function incorporates monetary rewards, which are ***another novel inclusion*** detailed in (3).
> 2. Not requiring data to be shared between devices and a central server. Data sharing is required in Karimireddy et al. (2022), our closest related mechanism, and it violates the true FL setting.
> 3. Not requiring design of a payment mechanism (like that in Contract Theory) which needs careful construction of the payment *prior to training*.
>     - This is similar to Karimireddy et al. (2022), however we instead define the central server's own utility and showcase that our mechanism maximizes the server's utility.
>     - Also unlike Karimireddy et al. (2022), we leverage our newly defined server utility to allow monetary rewards for participating devices. This is fully novel within the non-payment mechanism setting.
> 4. Demonstrating empirically (on real-world data) that RealFM succeeds in the goals laid out within our paper: server utility, device utility, and device contributions skyrocket while free-riding is negated when devices use RealFM.
>
> ### Data Quality Assumptions
>
> The overarching goal of our work is to create a foundational FL mechanism which solves many of the issues detailed in our paper as well as highlighted at the beginning of this rebuttal. There remain many frontiers to improve how realistically device utility is modeled, one of which is allowing for different quality data. Relaxing our assumptions on data quality is an important frontier to further improve realistic device utility. The literature [Wang et al. (2020)] provided by reviewer A7vi on data quality is a great place to begin tackling the data quality assumptions (through the use of Federated Shapley Values).
>
> > As the authors mentioned, it is hard to bond the composition function $\phi$, and I am still not sure how, particularly with this framework, we can ensure increased data production leads to better payoffs. In principle, one has to factor in the quality of data or limit combinatorial properties.
>
> Accuracy function $\hat{a}(m)$ is increasing with respect to the amount of data $m$ (due to its concavity). The accuracy-payoff function $\phi_i(a)$ is increasing with respect to accuracy $a$. Therefore, $\phi_i(a(m))$ is increasing with respect to $m$. This is how, given our assumptions on data quality, we ensure "increased data production leads to better payoffs".
>
> > Will it enjoy generalization to the non-i.i.d. case?
>
> Currently our mechanism does not fully generalize to the non-i.i.d case. The reason stems from the fact that our assumptions on $\hat{a}(m)$, detailed above, may fail in the non-i.i.d case ($\hat{a}(m)$ may not necessarily increase with more non-i.i.d data). However, there may be scenarios where adding non-i.i.d data still allows $\hat{a}(m)$ to be increasing with respect to $m$. Under this scenario our mechanism *would* generalize to the non-i.i.d case.

---

> ### Author Response · Authors · 2023-11-16
> **Reviewer gdSK Rebuttal (Continued)**
>
> We continue our rebuttal below and address the remaining questions within the review.
>
> ### Parameter Clarifications
>
> > Some inconsistencies and incompleteness due to unclear representation. For instance, the authors mention generalizing $\phi_i$ to become a convex and increasing function (but in which variable), and also, in actuality, the utility is modeled as a concave function? Following that, Assumption 2 needs further clarity. Another concern is Fig. 2, which is not well-justified for the said "payoff function. I don't see this claim has been experimentally validated apart from numerical evaluation.
>
> We provide clarification that **(1)** $\phi_i(a)$ is *convex* with respect to the accuracy $a$, however **(2)** $\phi_i(\hat{a}(m))$ must remain concave with respect to $m$.
>
> As detailed in our paper, we can define the accuracy function $\hat{a}(m)$ as either Equation 1 or 40 and the accuracy-payoff function $\phi_i(a) := \frac{1}{(1-a)^2} - 1$ and this will satisfy both **(1)** and **(2)**. There are other variants of $\hat{a}(m)$ and $\phi_i(a)$ which work, but these are backed up by generalization bounds (Proposition 1) and intuition respectively.
>
> The convex and increasing requirement for $\phi_i(a)$ is intuitive as rational devices heavily desire models with accuracies approaching 100%. This is much more intuitive than a linear $\phi_i(a)$, as described in the paper, since rational devices would place much greater value in a model increasing in accuracy from 99 to 99.5% than 30 to 30.5%.
>
> ---
>
> > How is $a_{opt}$ derived/obtained?
>
> In practice, $a_{opt}$ depends upon model architectures as well as the learning task at hand. In many cases, prior knowledge about the difficulty of the learning task can illuminate what an actual or approximate value of $a_{opt}$ should be (*e.g.,* many well-trained neural networks for image classification can achieve 90%+ optimal accuracies). In situations where the optimal accuracy might be completely unknown, a value close to 100% for $a_{opt}$ can be used to be conservative and ensure no issues will arise within the mechanism.
>
> ---
>
> > If the profit margin of the central server, defined as $p_m$, is fixed and known by all devices, I am not sure why it is used after all. (in the later experiments, this is used to indicate the degree to which the server is greedy). Assuming this is unknown would be an interesting analysis. Also, the definition of profit margin should be made more rigorous.
>
> The profit margin $p_m$ details the percent of utility kept by the server after federated training is complete. The remaining $(1-p_m)$ percent is distributed uniformly as a monetary reward to all participating devices. It is important for devices to know the profit margin $p_m$, since a smaller $p_m$ will result in larger rewards for each device and may provide further incentive to participate. As noted by the reviewer, we showcase that device utility and contributions increase *even when the server is greedy and maintains all of its gained utility*.
>
> We agree with the reviewer that assuming an unknown profit margin would be interesting analysis and we leave this as important future research. As detailed at the beginning of the rebuttal, our main goal was to construct the first FL mechanism of its kind. Once planted, we aim to further relax some of the assumptions initially used.

---

> > ### Comment · Reviewer_gdSK · 2023-11-20
> >
> > Many thanks authors for the response, and I'm sorry for the late feedback on this.
> > -  One concern was whether concavity would hold for $\hat{a}(m)$ with $m$, i.e., can the method generalize to the non-i.i.d. case? As the authors agreed, it does not, for instance, when having adversarial data samples. Then, it would be confusing when you say, "However, there may be scenarios where adding non-i.i.d data still allows to be increasing with respect to. Under this scenario, our mechanism would generalize to the non-i.i.d case". Further, this means the accuracy model might not hold practical without making strong (impractical) assumptions? I believe the authors should clarify such limitations in the manuscript.
> > -  I agree with Reviewers EigJ (and A7vi) for their concerns on simplified assumptions (as I raised in my earlier comment). For instance, the assumption that the closed form of each device's utility function is known may not always be realistic. The authors argued this is the first step towards realistic FL; however, it is also true the clients can bargain for incentives in a non-cooperative manner, where the assumptions for the utility model would be different, e.g., they could be private, just like in the auction-based methods, but truthful? This implies missing relevant works, as indicated by Reviewer A7vi.

---

> > > ### Author Response · Authors · 2023-11-20
> > > **gdSK Discussion**
> > >
> > > We want to thank reviewer gdSK for their response. Your discussion is much appreciated! Below we respond to your comments.
> > >
> > > ## Non-I.I.D Data
> > >
> > > > One concern was whether concavity would hold for $\hat{a}(m)$ with $m$, i.e., can the method generalize to the non-i.i.d. case? As the authors agreed, it does not, for instance, when having adversarial data samples... Further, this means the accuracy model might not hold practical without making strong (impractical) assumptions? I believe the authors should clarify such limitations in the manuscript.
> > >
> > > As mentioned in our paper, we assume data is i.i.d to *ensure that our assumptions on $\hat{a}(m)$ hold w.r.t $m$*. These are assumptions we look to relax in future work. **The main novelty of our work is, in the words of the reviewer, to create a FL mechanism which takes "the first step towards realistic FL".**
> > >
> > > > Then, it would be confusing when you say, "However, there may be scenarios where adding non-i.i.d data still allows to be increasing with respect to. Under this scenario, our mechanism would generalize to the non-i.i.d case".
> > >
> > > We want to clarify that our mechanism is not *guaranteed* to generalize to the non-i.i.d case. However, there do exist instances where $\hat{a}(m)$ remains concave w.r.t $m$ if data is slightly non-i.i.d [Sun et. al (2017)]. In these specific instances, since $\hat{a}(m)$ remains concave w.r.t $m$, our mechanism would still work. When data becomes highly non-i.i.d, our assumption breaks. Thus, we cannot provably guarantee our mechanism holds in the non-i.i.d case but there are instances where it will.
> > >
> > > ## Device Utility
> > >
> > > > Clients can bargain for incentives in a non-cooperative manner, where the assumptions for the utility model would be different, e.g., they could be private, just like in the auction-based methods, but truthful? This implies missing relevant works, as indicated by Reviewer A7vi.
> > >
> > >
> > > Indeed, our mechanism is equivalent to (and could be reformulated) as a private, yet honest, auction-based method. In this setting, each device will not share its explicit utility function with the server but will be truthful with its information. Likewise, the server will not share its utility function with any devices but will be truthful about its profit margin and final model accuracy.
> > >
> > >
> > > In the first phase of our mechanism, the phases are detailed in Figure 1, each device would truthfully share how much data it wants to contribute to the FL training process. In the second phase, federated training occurs. Once complete, we reach the third and final phase: reward distribution. Instead of providing the server with each device's cost and accuracy-payoff function, the server can truthfully share the final model accuracy and marginal reward per data point contributed with each device. In turn, each device will truthfully compute the amount of accuracy it should receive based on its accuracy-shaping function $\gamma_i$ and subsequently query the server for a model with this prescribed accuracy.
> > >
> > > The entire process laid out above is equivalent to Algorithm 1 within our paper. The difference is that in Algorithm 1 devices share their information with the server whereas in the process above the server shares its information with each device.
> > >
> > > **Remark:** We will add the discussion about this auction-method reformulation into our revised paper. Finally, we note that we covered the relevant works detailed by reviewer A7vi in our revised paper within Appendix A. Thank you for raising this point.
> > >
> > > ### References
> > > Sun et. al, Revisiting Unreasonable Effectiveness of Data in Deep Learning Era, 2017.

---

> > > > ### Comment · Reviewer_gdSK · 2023-11-22
> > > >
> > > > I thank the authors for their valuable efforts in clarifying the concerns I raised before. Actually, if I can kindly disagree, I wrote, "The authors argued this is the first step towards realistic FL", rather than what you have mentioned before in your response: $\textbf{The main novelty of our work is, in the words of the reviewer, to create a FL mechanism which takes "the first step towards realistic FL"}.$ I am still not sure, with i.i.d., assumption, the accuracy model is a "realistic step". Thank you for clarifying that it cannot generalize to non-i.i.d. cases but covers only a few cases, if so.
> > > >
> > > > Leaving that aside, as we are at the final day of the discussion period, considering the merit and contribution of this work and following up on other reviewers' comments and your responses to them, I am keeping my scores at this stage.

---

> > > > > ### Author Response · Authors · 2023-11-22
> > > > > **gdSK Discussion**
> > > > >
> > > > > Thank you for your follow-up replies. We are happy to hear that our previous replies clarified the concerns you raised before.
> > > > >
> > > > > > Actually, if I can kindly disagree, I wrote, "The authors argued this is the first step towards realistic FL", rather than what you have mentioned before in your response
> > > > >
> > > > > We apologize for our misquote, and recognize that you were simply quoting what we had mentioned in our previous rebuttals.
> > > > >
> > > > > > I am still not sure, with i.i.d., assumption, the accuracy model is a "realistic step".
> > > > >
> > > > > We agree that relaxing the i.i.d data assumption will further improve our mechanism. At the same time, *we want to emphasize that the major novelties of our paper lie within*:
> > > > > 1. Designing a non-linear device utility function,
> > > > > 2. Creating an accuracy-shaping incentive, with tight guarantees, which provably fosters device participation and contributions,
> > > > > 3. Relaxing assumptions of data sharing compared to previous related literature,
> > > > > 4. Designing a payment mechanism which does not need to be put into effect prior to training,
> > > > > 5. Demonstrating the success of our mechanism empirically on real-world data towards improving overall utility and contributions.
> > > > >
> > > > > We believe that these contributions are both novel and important to the field of mechanism design in FL. Thank you again reviewer gdSK for your discussion. Let us know if any further questions remain before the end of the discussion period.

---

### Official Review · Reviewer_EigJ · 2023-10-30

**Soundness:** 3 good
**Presentation:** 2 fair
**Contribution:** 2 fair
**Rating:** 3
**Confidence:** 4

**Summary:**

To address the free-rider issue in federating learning, this paper proposes RealFM, a mechanism that takes into account device utility and incentivizes data contribution and device participation. Compared with previous work by Karimireddy et al. (2022), RealFM allows for a non-linear relationship between device accuracy and utility.

**Strengths:**

1. The paper presents an interesting approach to address the free-rider problem in federated learning.
2. Convex utility functions do hold in many instances, and it is necessary to design a mechanism to incentivize device participation for such cases, as this paper does.

**Weaknesses:**

1. The assumption that the closed form of each device's utility function is known, which may not always be realistic.
2. RealFM distributes model accuracy and rewards based on the amount of data provided by each device. This may raise questions about potential cheating. It would make more sense to depend on each device's marginal contribution to model accuracy, provided that it can be easily and accurately determined.
3. Theorem 4, which claims that RealFM eliminates the free-rider phenomena, may not be particularly thrilling. Designing a contract to incentivize individuals when their exact contribution is known is not a challenging task.
4. The comparison between linear RealFM and local training does not effectively demonstrate how well RealFM incentivizes non-linear devices' contribution to model training. It is understandable that the authors cannot be blamed for weak baseline algorithms, as this paper is the first to aim at incentivizing device data contribution in a non-linear setting. However, in such cases, it would be more valuable to have theoretical guarantees of data maximization to truly showcase the strength of RealFM, which the paper lacks, except for the linear setting.

**Questions:**

The authors provide explanations for modeling the relationship between model accuracy and data quantity as Eq. (1) and (2), it would be interesting to explore whether the results can be generalized to accommodate more general accuracy functions.

---

> ### Author Response · Authors · 2023-11-15
> **Reviewer EigJ Rebuttal**
>
> We thank reviewer EigJ for their thoughtful review of our paper. Below we address the questions raised.
>
> > The assumption that the closed form of each device's utility function is known, which may not always be realistic.
>
> We agree with the reviewer that assuming the form of $u_i$ can be unrealistic in some instances for each device. However, we believe that using a closed-form $u_i$ is an important first step towards realistic FL. Using a closed-form $u_i$ for each device is much more realistic than assuming that devices will always participate in federated training regardless of its costs. The majority of FL papers do not even attempt to model device or server utility, which is quite unrealistic. Furthermore, there is usage of closed-form utility functions in economics literature [Trout (1963) & Orsborn et al. (2009)] as well as with mechanism design in ML [Karimireddy et al. (2022) and Zhan et al. (2021; 2020b)] to name a few. Removing the closed-form assumption is a great future research direction.
>
> > This may raise questions about potential cheating. It would make more sense to depend on each device's marginal contribution to model accuracy, provided that it can be easily and accurately determined.
>
> We are wary that many other avenues to determine each device's marginal contribution, like Federated Shapley Value Wang et al. (2020), can be cheated as well. The issues surrounding honesty are an *important* future research direction in FL. We set out to accomplish quite a bit in our work, as **it is the first FL mechanism which more realistically models device utility (non-linear function of model accuracy) and provably improves device participation and contributions (eliminating the free-rider effect) without the need of a payment mechanism**. Specific novelties are listed in our rebuttal for Reviewer A7vi.
>
> In summary, we aim to address honesty in future works but we first wanted to establish a foundational base for FL mechanisms which incentivize device participation and contributions.
>
> > Theorem 4, which claims that RealFM eliminates the free-rider phenomena, may not be particularly thrilling. Designing a contract to incentivize individuals when their exact contribution is known is not a challenging task.
>
> As mentioned within our paper, contract-theory approaches require construction of a payment mechanism. Pricing of the contracts is especially difficult and can often result in sub-optimal utility gained by both the devices and servers (too low of a reward does not incentivize devices to participate and too high of a reward can cause the server to attain low or negative utility).
>
> The importance of our work is that our mechanism allows free-riding to be eliminated **without having to construct a payment mechanism prior to training**. Using accuracy shaping, devices are incentivized to contribute more data because they will achieve a higher-performing model than they would by training alone. Theory 4 provably eliminates free-riding while also guaranteeing that devices will *increase* their contributions under a new accuracy-shaping scheme, differing from Karimireddy et al. (2022), which enables flexible modeling of diverse device utilities that accommodate non-linear relationships between model accuracy and utility. Furthermore, this mechanisms allow for monetary rewards to be received by devices after training.

---

> ### Author Response · Authors · 2023-11-15
> **Reviewer EigJ Rebuttal (Continued)**
>
> We continue our rebuttal below and address the remaining questions within the review.
>
> > It is understandable that the authors cannot be blamed for weak baseline algorithms, as this paper is the first to aim at incentivizing device data contribution in a non-linear setting... it would be more valuable to have theoretical guarantees of data maximization to truly showcase the strength of RealFM, which the paper lacks, except for the linear setting.
>
> As mentioned by the reviewer, due to the dearth of FL mechanisms that allow for non-linear relationships between model accuracy and utility we were unable to compare against other mechanisms. **We believe this speaks volumes about the novelty and importance of our work**.
>
> As mentioned by the reviwer, we prove data maximization under linear settings in Corollary 1. We were unable to guarantee tight theoretical guarantees in other settings due to the non-linearity. This stems from the proof of Theorem 3, specifically relaxing Equation 29. In fact, we assume $\phi_i(a)$ is convex with respect to the accuracy $a$ **in order to make the bound as tight as possible**. Therefore, the accuracy-shaping function proposed in Theorem 3 is close to data maximizing but falls slightly short.
>
> We also mention that our empirical results showcase major improvement (up to 7x more) in device contributions relative to the available baselines even when the profit margin is set to 100%. Our results are even stronger when the server is not completely greedy. *We outperform all state-of-the-art mechanisms which is both novel and important.*
>
> > ...it would be interesting to explore whether the results can be generalized to accommodate more general accuracy functions.
>
> As long as the accuracy function $a$ fulfills the mild and reasonable assumptions laid out in Assumption 1: continuous, non-decreasing, and concave (along a certain interval) with respect to the amount of data $m$. These requirements are intuitive and backed by empirical studies Sun et al. (2017): accuracy grows with the amount of data used (continuous and non-decreasing) yet experiences diminishing returns (concave).

---

### Official Review · Reviewer_A7vi · 2023-10-30

**Soundness:** 2 fair
**Presentation:** 3 good
**Contribution:** 2 fair
**Rating:** 5
**Confidence:** 4

**Summary:**

This paper proposes RealFM to incentivise devices to participate in training and alleviate the threat of free riders. The paper considers that each rational device wish to maximise its utility, which depends on its cost of participation and non-linearly on its model accuracy and monetary reward. RealFM involves giving each device a monetary reward and design/use a “accuracy shaping”  function to boost a device’s model accuracy and incentivize it to produce more data.

**Strengths:**

- The paper novelly consider utility as a non-linear function of model accuracy. This makes accuracy shaping (incentivising contribution beyond local optimal amount) harder.
- The paper is generally written well although some notations should be defined earlier.

**Weaknesses:**

- Some assumptions are simplistic and limit the significance of the work. In equation 1, the accuracy is modelled to depend solely on the number of data points but in practice, devices may have data with different quality, diversity and noise. Moreover, it is hard to decide the difficulty of the learning task $k$ and the server would not have access to data for tuning (Sec. 6) before incentivization. As $a$ is an upper bound, the individual rationality for the upper bound does not translate to rationality for the expected/actual values.
- The advantage of this work over the existing work has to be made clearer. There needs to be a deeper discussion about Shapley value based/collaborative fairness approaches including Xu et. al. (2021) and Wang et. al. (2020). In appendix A, it is suggested that these work “assume that devices are already willing to contribute all of their data”. This claim may be inaccurate — these work guarantees that contributing less (nothing) leads to less (no) reward thus devices would respond by contributing more data as in this work. The difference is that there is no closed form function for device $i$ utility.

  Wang, T., Rausch, J., Zhang, C., Jia, R., & Song, D. (2020). A principled approach to data valuation for federated learning. Federated Learning: Privacy and Incentive, 153-167.
- Some notations are used before they are defined or explained (e.g., $[\mathcal{M}^U(\cdot)]_i $ in Theorem 2). This makes the claims unclear and harder to understand.
- The central server has to compute $m_i^o$ and $m_i^*$ based on the declared $\phi_i$ and $c_i$. Device $i$ may misreport the cost to get a better reward.

**Questions:**

Questions
1. In related works, it is mentioned that Karimireddy et al. (2022) “requires data sharing between devices and the central server”. Is the amount of data sharing the same as centralised FL? How does the amount of sharing in this work differ?
2. The Shapley value and collaborative fairness based approaches in existing work may not assume that devices are willing to contribute data. They just guarantee if they are unwilling and contribute less (nothing), they receive less (no) reward. Resultingly, rational devices should contribute more data as in this work. Can you clarify and make the differences/advantages of your work more specific?
3. What is the implication/significance of Theorem 2?
4. Does the mechanism _need_ monetary rewards to incentivise devices (can the profit margin be set to 1)?
5. In Eq 10, must $\phi_C(a(\sum m))$ be less than $\sum m$ to ensure a profit margin?
6. What does $\epsilon$ in Theorem 3 represent or control?
7. In practice, how does the central server produce a model with accuracy $a(m_i^o) + \gamma_i(m_i)$? Is it guaranteed to be less than $a(\sum m)$? Does the server solve for the number of additional data points to use?


Minor suggestions
* The theorem and corollary should come with intuitive description to aid understanding. For example, for C1, a device with lower marginal cost has higher utility and would contribute more data.
* In definition 1, $m$ can be used in place of $\sum \mathbf{m}$. $\mathbf{m}$ is not defined.

---

> ### Author Response · Authors · 2023-11-15
> **Reviewer A7vi Rebuttal (Weaknesses)**
>
> We thank reviewer A7vi for their thoughtful review of our paper. Below we address the questions raised. We first address the "Weakness" section within the review. Equations referenced below correspond to our revised paper.
> ### Novelty (Advantages) of our Work
> We begin our rebuttal by detailing the novelty of our work. **Our work is the first FL mechanism which more realistically models device utility (non-linear function of model accuracy) and provably improves device participation and contributions (eliminating the free-rider effect) without the need of a payment mechanism prior to training**. We sought to design a foundational FL mechanism which improves upon previous FL mechanisms by:
> 1. Modeling device utility in a more realistic manner which enables flexible modeling of diverse device utilities that accommodate non-linear relationships between model accuracy and utility.
>      - Allowing this non-linear relationship within device utility ***requires major changes*** to the accuracy-shaping procedure, which incentivizes increased contributions in exchange for higher model accuracy, at the heart of the incentive mechanism (Theorem 4).
>      - The altered accuracy-shaping function and its provable guarantees in Theorem 3 ***are novel***.
>      - The accuracy-shaping function incorporates monetary rewards, which are ***another novel inclusion*** detailed in (3).
> 2. Not requiring data to be shared between devices and a central server. Data sharing is required in Karimireddy et al. (2022), our closest related mechanism, and it violates the true FL setting.
> 3. Not requiring design of a payment mechanism (like that in Contract Theory) which needs careful construction of the payment *prior to training*.
>     - This is similar to Karimireddy et al. (2022), however we instead define the central server's own utility and showcase that our mechanism maximizes the server's utility.
>     - Also unlike Karimireddy et al. (2022), we leverage our newly defined server utility to allow monetary rewards for participating devices. This is fully novel within the non-payment mechanism setting.
> 4. Demonstrating empirically (on real-world data) that RealFM succeeds in the goals laid out within our paper: server utility, device utility, and device contributions skyrocket while free-riding is negated when devices use RealFM.
> ---
> ### Simplicity of Assumptions
> > (1) Accuracy is modeled to depend solely on the number of data points... devices may have data with different quality, diversity and noise.
> > (2) The central server has to compute $m_i^o$ and $m_i^*$ based on the declared $\phi_i$ and $c_i$. Device $i$ may misreport the cost to get a better reward.
>
> The overarching goal of our work is to create a foundational FL mechanism which solves many of the issues detailed in our paper as well as above. There remain many frontiers to improve how realistically device utility is modeled. Data and truthfulness are two of them. We agree that in practice devices have different data distributions and data quality can vary both inter- and intra-device. We also agree that devices can be dishonest and report untruthful costs in practice. Relaxing our assumptions on data quality and truthfulness is an important follow-up to our work. The literature provided by reviewer A7vi on data quality is a great place to begin tackling the data quality assumptions.
>
> > Hard to decide the difficulty of the learning task $k$ and the server would not have access to data for tuning (Sec. 6) before incentivization. As $a$ is an upper bound, the individual rationality for the upper bound does not translate to rationality for the expected/actual values.
>
> The use of $a$ in our paper was, in part, to allow for empirical analysis of our mechanism. What is most important about $a$ is that it must hold under reasonable and mild assumptions: continuous, non-decreasing, and concave (along a certain interval) with respect to the amount of data $m$. As shown in empirical studies Sun et al. (2017), accuracy grows with the amount of data used (continuous and non-decreasing) yet the growth experiences diminishing returns (concave). In practice, $a$ can be tuned by the server to align with test accuracy curves (as the accuracy improves with more contributions and updates).
>
> ---
> ### Paper Refinements
> > (1) There needs to be a deeper discussion about Shapley value based/collaborative fairness approaches including Xu et. al. (2021) and Wang et. al. (2020)
> > (2) Some notations are used before they are defined or explained (e.g., $[\mathcal{M}^U(\cdot)]_i$ in Theorem 2). This makes the claims unclear and harder to understand.
>
> We appreciate the review's feedback and have expanded our related works and fixed our mistaken claim in Appendix A. $[\mathcal{M}^U(\cdot)]_i$ is the utility received by a device $i$ participating in the mechanism. We added or moved definitions of notations to ensure they are defined when they first appear. We have fixed all minor suggestions.

---

> ### Author Response · Authors · 2023-11-15
> **Reviewer A7vi Rebuttal (Questions)**
>
> We continue our rebuttal here and address the "Questions" section within the review.
>
> > Karimireddy et al. (2022) “requires data sharing between devices and the central server”. Is the amount of data sharing the same as centralised FL? How does the amount of sharing in this work differ?
>
> No data is shared within centralized FL. Instead, gradient updates are communicated between the devices and a central server. In Karimireddy et al. (2022), devices share their actual data with a server *which violates the fundamental underpinning of FL*.
>
> > Shapley value and collaborative fairness based approaches in existing work may not assume that devices are willing to contribute data... rational devices should contribute more data as in this work. Can you clarify and make the differences/advantages of your work more specific?
>
> We detail the novelty of our work in the first part of our rebuttal. Wang et al. (2020) proposes a method to estimate the federated Shapley Value (which holds similar properties as the regular Shapley Value) in order to determine data quality from the devices participating in FL training. Thus, Wang et al. (2020) and other Shapley Value works seek to evaluate each data source in the FL setting. *Our work seeks to incentivize and increase device participation, through mechanism design, within FL. This is fundamentally different.*
>
> Collaborative fairness works such as Lyu et al. (2020) are closer to our own in that they seek to fairly allocate models with varying performance depending upon how much devices contribute to training in FL settings. There are a few **key** differences between this line of work and our own, namely:
> 1. Our mechanism incentivizes devices to both participate in training and increase their amount of contributions. There are no such incentives in collaborative fairness.
> 2. We model device and server utility. Unlike collaborative fairness, we do not assume that devices will always participate in FL training.
> 3. Our mechanism design **provably** eliminates the free-rider phenomena unlike collaborative fairness, since devices who try to free-ride receive the same model performance as they would on their own (Equations 12 & 13).
> 4. No monetary rewards are received by devices in current collaborative fairness methods.
>
> > What is the implication/significance of Theorem 2?
>
> Theorem 2 provides the theoretical underpinnings for a mechanism with certain assumptions to reach a Nash Equilibrium. Our mechanism satisfy these assumptions and thus reach an equilibrium at which devices will not deviate from their strategies (contributions). We prove in Theorem 3 that accuracy shaping used within our mechanism provably increases device utility and contributions. By Theorem 4, our mechanism reaches an equilibrium (Theorem 2) at which devices contribute more, receive higher utility, and no free-riding exists (Theorem 3).
>
> > Does the mechanism need monetary rewards to incentivise devices (can the profit margin be set to 1)?
>
> No. Monetary rewards are not required. In fact, for ease of empirical analysis, *we set the profit margin equal to 1 in all of our experimental results*. Thus, device contributions and utility would **increase** further if we had set the profit margin to a lower value. This showcases how strong our mechanism works empirically.
>
> > In Eq 10, must $\phi_C(a(\sum m))$ be less than $\sum m$ to ensure a profit margin?
>
> There may be some confusion regarding the profit margin. The profit margin $p_m$ dictates what *percentage* of the overall utility gained by the server is kept by the server. The remaining $1-p_m$ percentage of the utility is uniformly returned (as a marginal monetary reward) to each participating device as defined in Equation 10. Overall, $\phi_C(a(\sum m))$ can be larger, equal to, or smaller than $\sum m$. This does not affect the profit margin, it only affects the amount of monetary rewards returned to each participating device.
>
> > What does $\epsilon$ in Theorem 3 represent or control?
>
> The parameter $\epsilon$ simply ensures that Equation 11 will be a **strict** inequality. This is crucial to ensure that devices are receiving more utility participating in our mechanism, via accuracy shaping, than on their own. If the inequality is not strict then devices are not incentivized to increase their contributions.
>
> > In practice, how does the central server produce a model with accuracy $a(m_i^o) + \gamma_i(m_i)$? Is it guaranteed to be less than $a(\sum m)$?
>
> Via FL, the server will train a model using all contributions $\sum m$ which will result in a model with accuracy $a(\sum m)$. In order to provide devices with lower model performance, such as $a(m_i^o) + \gamma_i(m_i)$, the server can degrade the model using a variety of methods. Some examples of ways to do this include using noisy perturbations in a controlled manner or returning a model midway through training which has lower performance.

---

> > ### Comment · Reviewer_A7vi · 2023-11-21
> >
> > Thanks for the response and clarifications that addressed my questions and improved the work! Here are some additional concerns.
> >
> > > In practice, $a$ can be tuned by the server to align with test accuracy curves (as the accuracy improves with more contributions and updates).
> >
> > The server would need data to train models and obtain the test accuracy curves. However, it may be problematic that the server may lack data to do so before incentivizing contribution.
> >
> > > 1. Our mechanism incentivizes devices to both participate in training and increase their amount of contributions. There are no such incentives in collaborative fairness.
> > 3. Our mechanism design provably eliminates the free-rider phenomena unlike collaborative fairness, since devices who try to free-ride receive the same model performance as they would on their own (Equations 12 & 13).
> > 4. No monetary rewards are received by devices in current collaborative fairness methods.
> >
> > The fairness axioms satisfied by the Shapley value may incentivise devices to increase their amount of contributions and ensure that a null player gets no reward. As an example, see the uselessness and strict monotonicity property in Sim et al. 2020.
> >  Collaborative fairness and data valuation works include a "data valuation" component that can be used to decide monetary reward (Jia, R., Dao, D., Wang, B., Hubis, F.A., Hynes, N., Gürel, N.M., Li, B., Zhang, C., Song, D. and Spanos, C.J., 2019, April. Towards efficient data valuation based on the shapley value. In The 22nd International Conference on Artificial Intelligence and Statistics (pp. 1167-1176). PMLR.)
> >
> > > No. Monetary rewards are not required. In fact, for ease of empirical analysis, we set the profit margin equal to 1 in all of our experimental results. Thus, device contributions and utility would increase further if we had set the profit margin to a lower value. This showcases how strong our mechanism works empirically.
> >
> > The inclusion of the monetary rewards seems a bit unnecessary and separate from other contributions of this work. Can the authors highlight the importance of monetary rewards (in addition to accuracy rewards) and the challenges of designing it?

---

> > > ### Author Response · Authors · 2023-11-22
> > > **A7vi Discussion**
> > >
> > > We want to thank reviewer A7vi for their response. Your discussion is much appreciated! Below we respond to your comments.
> > >
> > > > The server would need data to train models and obtain the test accuracy curves. However, it may be problematic that the server may lack data to do so before incentivizing contribution.
> > >
> > > In practice, our mechanism only needs to know the accuracy function $a(m)$ within the domain [0, $\sum \textbf{m}$]. The reason is that accuracy shaping only works up until the final fully trained model accuracy. It is not possible to provide more accuracy than $a(\sum \textbf{m})$, as this would render the mechanism infeasible. Thus, the accuracy curve obtained by the server through training is enough to incentivize contribution.
> > >
> > > ***Note on Novelty:*** We want to note that the ability for our mechanism to distribute rewards after training is another novelty to our work compared to previous mechanisms in FL. We can ensure individual rationality before contributions are distributed or training is performed!
> > >
> > > > Fairness axioms... incentivise devices to increase their amount of contributions and ensure that a null player gets no reward. As an example, see the uselessness and strict monotonicity property in Sim et al. 2020.
> > >
> > > ***Contrast with Sim et al. (2020):*** We first note that Sim et al. (2020) assumes a data sharing environment which is not within the FL setting. As such, these fairness axioms may no longer hold within FL. Second, the collaborative learning frameworks seek to fairly allocate model performance in a distributed setting. However, increasing device participation is not a goal. ***Our paper is fundamentally different, as we seek to design a mechanism which a server can adopt to incentivize and increase both device participation and contributions.***
> > >
> > > With that being said, we *greatly* appreciate the papers which you have provided us. They illuminate how our monetary rewards, as well as accuracy rewards, can be further optimized and made more realistic when incorporating the true quality of data. This is a crucial future research direction. We again want to make clear that the novelty and goal of our work is to design the first FL mechanism which more realistically models device utility (non-linear function of model accuracy) and provably improves device participation and contributions (eliminating the free-rider effect) without the need of a payment mechanism prior to training. We hope to add a data valuation mechanism in future work.
> > >
> > > ---
> > >
> > > ## Monetary & Accuracy Reward Importance and Design Challenges
> > >
> > > **(Monetary Reward Importance)** We would like to clarify that providing monetary rewards is quite important within our work, due to its relationship with accuracy shaping as seen in Theorem 3. Providing monetary rewards allows devices to make up the cost of collecting more data $(c_i - r(\textbf{m}))$. The effect of this allows our incentive mechanism to stretch the domain of accuracy shaping, incentivizing devices to contribute even more than if no rewards were distributed.
> > >
> > > **(Accuracy Reward Importance)** Accuracy reward is the main driver of incentive in our mechanism. Devices want to gain access to the fully trained model $a(\sum \textbf{m})$, and can receive it in exchange for providing more contributions than locally optimal. This is what takes place within the accuracy shaping function.
> > >
> > > **(Design Challenges)** While modeling device utility in a manner which allows a non-linear relationship between model accuracy and utility is more realistic than previous works, it came with many challenges. One of the largest challenges was constructing a **novel** accuracy shaping function which can account for a non-linear accuracy-payoff function $\phi_i$. In Karimireddy et al. (2022), the accuracy function was a simple linear function as detailed in Remark 1. Moving to the more realistic device utility forced the construction of a non-linear accuracy-shaping function $\gamma_i$. This accuracy-shaping function ensures that devices receive marginally more utility gained from the improved accuracy than utility lost by contributing more data (Individual Rationality). We carefully construct $\gamma_i$, and place convexity assumptions on $\phi_i$, in order to ensure our bound on data maximization is as tight as possible.
> > >
> > > The monetary reward design faced additional challenges as our FL mechanism is the first of its kind (*i.e., using accuracy shaping*) to design server utility and allow for a payment mechanism which takes place after training finishes (no contract required prior to training). We determined that distributing uniform marginal monetary rewards amongst participating devices (each device $i$ receives $r(\textbf{m})\cdot m_i$ reward) allows us to improve the domain of accuracy-shaping for *all participating devices*. These novelties allow us to improve and strengthen the amount of data maximization that our mechanism provides to the server even under more realistic device utilities.

---

> > > > ### Comment · Reviewer_A7vi · 2023-11-23
> > > >
> > > > Is it correct that the accuracy function $a$ is only needed after FL training is completed (Step 7 of Algo 1)? Hence, the server can train multiple models to estimate $a(m)$ for any $m \in [0, \sum \mathbf{m}]$ during training?
> > > >
> > > > For the last question, I meant why is monetary reward important and defined as in Eq 9 when the server is already giving model/accuracy reward. The response on (Monetary Reward Importance) partially address my question.
> > > > Can you clarify and elaborate the impact of decrease in the profit margin $p_m$ on $\gamma_i$?
> > > > When $p_m$ is lower, $r(\textbf{m})$ is higher, so $c_i - r(\textbf{m})$ and $\gamma_i$ (the additionally utility) is lower?

---

> > > > > ### Author Response · Authors · 2023-11-23
> > > > >
> > > > > > Is it correct that the accuracy function is only needed after FL training is completed (Step 7 of Algo 1)? Hence, the server can train multiple models to estimate $a$ for any $m \in [0, \sum \textbf{m}]$ during training?
> > > > >
> > > > >  Yes, this is correct!
> > > > >
> > > > >  > For the last question, I meant why is monetary reward important and defined as in Eq 9 when the server is already giving model/accuracy reward. The response on (Monetary Reward Importance) partially address my question. Can you clarify and elaborate the impact of decrease in the profit margin $p_m$ on $\gamma_i$? When $p_m$ is lower, $r(\textbf{m})$ is higher, so $c_i - r(\textbf{m})$ and $\gamma_i$ (the additionally utility) is lower?
> > > > >
> > > > > We first want to clarify that $\gamma_i$ is the additional *accuracy* given to each device that contributes more to the server than is locally optimal. In Equation 11, we prove that our novel construction of $\gamma_i$ (the additional amount of accuracy provided via accuracy shaping) ensures that devices receive positive utility for contributing more data.
> > > > >
> > > > > When $p_m$ is lower, the server will keep less of the utility it gains and spreads the (larger) remaining utility uniformly as a monetary reward $r(\textbf{m})$ amongst participating devices. This monetary reward counteracts the cost that devices incur for collecting and contributing more data past its locally optimal amount $(c_i - r(\textbf{m})) \cdot m_i$. As this cost is lowered, the added accuracy $\gamma_i$ can be lower since a smaller cost $c_i - r(\textbf{m})$ is incurred. This is detailed explicitly in Equation 11:
> > > > > $$
> > > > > \phi_i(a(m^o_i
> > > > > ) + \gamma_i(m)) − \phi_i(a(m^o_i)) > (c_i − r(\textbf{m}))(m − m^o_i) \quad \forall m \geq m^o_i.
> > > > > $$
> > > > > A lower profit margin increases $r(\textbf{m})$ which in turn lowers the right hand side of Equation 11 and allows $\gamma_i$ to be smaller. The effect of a smaller $\gamma_i$ is that $\gamma_i$ can span a larger domain until it meets the limit of accuracy shaping. This results in a larger optimal contribution for participating devices $m_i^*$:
> > > > > $$
> > > > > m_i^* := [ m \geq m_i^o  |  a(m + \sum_{j \neq i} m_j) = a(m^o_i) + \gamma_i(m) ].
> > > > > $$

---

> > > > > > ### Comment · Reviewer_A7vi · 2023-11-23
> > > > > >
> > > > > > > The effect of a smaller $\gamma_i$ is that  $\gamma_i$ can span a larger domain until it meets the limit of accuracy shaping. This results in a larger optimal contribution $m_i^*$ for participating devices
> > > > > >
> > > > > > Can you clarify this further?

---

### Official Review · Reviewer_jpGm · 2023-10-31

**Soundness:** 3 good
**Presentation:** 3 good
**Contribution:** 3 good
**Rating:** 6
**Confidence:** 3

**Summary:**

The presented research addresses the challenges of edge device participation in federated learning (FL) and the shortcomings of existing FL frameworks when applied in real-world contexts, particularly in addressing the free-rider problem. In response to these issues, the authors propose a novel approach called RealFM, which introduces several key innovations including Realistic Device Utility Modeling, Incentivizing Data Contribution and Participation, and Elimination of the Free-Rider Phenomenon. Experiments show that RealFM exhibits excellent performance.

**Strengths:**

RealFM represents a noteworthy contribution to the field of federated learning, addressing the need for more realistic settings and incentives for edge device participation. Its ability to model device utility, eliminate the free-rider problem, and improve utility and data contribution is particularly promising for advancing FL in real-world applications.

**Weaknesses:**

1. In the experimental setting, the number of devices is not large, which is not very consistent with the actual application scenario.
2. Intuitively I know roughly what Server Utility and Average Data Contribution mean. But in detail, I may still not fully understand how Server Utility and Average Data Contribution are Numerized. In particular, I don’t quite understand why Server Utility has been improved so much.

**Questions:**

How will performance change when the number of devices increases?
Can you explain Server Utility and Average Data Contribution in simpler and more intuitive language?

---

> ### Author Response · Authors · 2023-11-16
> **Reviewer jpGm Rebuttal**
>
> We thank reviewer jpGm for their thoughtful review and look forward to our discussion. Below we address the questions raised in their review.
>
> ### Experimental Size
>
> We want to note that our experiments were run in a truly parallel and federated setting. Within our compute cluster, we assigned each CPU/process as a device and utilized MPI to perform communication between CPUs and a server CPU. Furthermore, each CPU was pinned to a GPU on which gradient computations were performed.
>
> > In the experimental setting, the number of devices is not large, which is not very consistent with the actual application scenario.
>
> Due to the amount of compute resources, we could only use 16 CPUs and 8 GPUs (2 CPUs pinned to 1 GPU). We do lack the resources to perform realistic federated training in the realms of hundreds or thousands of devices. Furthermore, many existing FL papers use a similar scale as our own. Finally, other FL mechanism papers either use a similar number of devices *or do not perform truly federated experiments at all*. Our work is one of the first to provide truly federated empirical evidence to back up the claims of our proposed mechanism.
>
> We also lack sufficient amounts of data to be split amongst devices if we have thousands of devices. The number of unique samples in the MNIST dataset is 60k, and this is even smaller since a fraction needs to be removed to evaluate test accuracy. In future work, with the proper compute power, we would love to showcase the efficacy of our mechanism on thousands of devices with extremely large real-world data.
>
> > How will performance change when the number of devices increases?
>
> Our mechanism should improve as the number of participating devices increases. The reason is that more devices means more data used during FL training. Through the mild assumptions on $a(m)$, increasing and concave, more overall data contributions $\mathbf{m}$ will result in higher final accuracies $a(\sum\mathbf{m})$. The higher final accuracy will create a larger window for accuracy shaping, $m_i^* := [ m ≥ m^o_i | a(m + \sum_{j \neq i} m_j) = a(m^o_i) + \gamma_i(m) ]$, which in turn will result in a stronger incentive for devices to contribute more.
>
> ### Utility Explanation
>
> > Can you explain Server Utility and Average Data Contribution in simpler and more intuitive language?
>
> Server utility is simply the net benefit gained by the server from having a trained model with accuracy $a$: $\phi_C(a)$. Within our paper, the accuracy is a function of the total data contributions $\mathbf{m}$, and thus the final server utility is $p_m \cdot \phi_C(a(\sum \mathbf{m}))$. $p_m$ is just the profit margin, or the percentage of utility kept by the server after federated training is complete. The remaining $(1-p_m)$ percent is distributed uniformly as a monetary reward to all participating devices.
>
> Average data contribution is the average amount of data that each device has determined is optimal (*i.e.,* when device utility is maximized) to contribute. We see experimentally that the average amount of data contributions when devices train locally is substantially smaller than the optimal amount when devices participate in our mechanism. Furthermore, the average device utility also dramatically increases when participating in our mechanism!
>
> > I may still not fully understand how Server Utility and Average Data Contribution are Numerized.
>
> Following literature in economics, mechanism design, and the intersection of both in FL, utility is often defined as a real-valued number. This number denotes the "net benefit" that a device attains. A larger real value denotes a larger net benefit received by the device.
>
> > In particular, I don’t quite understand why Server Utility has been improved so much.
>
> As detailed above, server utility drastically improves because our mechanism incentivizes devices to contribute more data. By contributing more data, the accuracy $a(\sum \mathbf{m})$ increases (due to our mild assumptions) which in turn increases server utility $\phi_C(a(\sum \mathbf{m}))$ since $\phi_C$ is also strictly increasing with respect to the accuracy $a$.

---

> > ### Comment · Reviewer_jpGm · 2023-11-23
> > **Official Comment by Reviewer jpGm**
> >
> > Thanks for the detailed response. I have read the rebuttal and most of my concerns have been well addressed. Overall, I am towards acceptance and will maintain my score.
> >
> > Best, The reviewer jpGm

---

### Author Response · Authors · 2023-11-20
**Continued Discussion (Before End of Discussion Period)**

Dear Reviewers,

Thank you so much for your time and efforts in reviewing our paper. We have addressed your comments in detail and are happy to discuss more if there are any additional concerns. We are looking forward to your feedback.

Thank you,

Authors

---

### Meta-Review · Area_Chair_5QN7 · 2023-12-11

**Metareview:**

Thank you for your submission and your active discussion with the reviewers. The paper studies a new approach to incentivize device participation in federated learning (FL). The reviewers appreciated the departure from linear utility as a novelty. However, the reviewers also raised some concerns about the paper that were not fully resolved during the discussion period. The paper seems to rely on rather strong assumptions, including that model accuracy is only a function of the number of examples. There were also concerns about whether the proposed mechanisms prevent cheating behavior. Relatedly, perhaps the authors should consider a mechanism design perspective of the problem, where users may not truthfully report their number of data points.

**Justification For Why Not Higher Score:**

Despite the active discussion, the results still do not meet the acceptance bar.

**Justification For Why Not Lower Score:**

N/A

---

### Decision · Program_Chairs · 2024-01-16

Reject